# Don't Scale It All: Training-Free Localized Test-Time Scaling for Diffusion Models

## Abstract

Diffusion models have become the core paradigm for high-fidelity image generation, achieving remarkable performance in tasks such as text-to-image synthesis. A common strategy to further boost their performance is *test-time scaling* (TTS), which improves generation quality by allocating more computation during inference. Despite recent progress, existing TTS methods operate at the full-image level, neglecting the fact that image quality is often spatially heterogeneous. As a result, they squander computation on already satisfactory regions while failing to target localized defects, leading to both inefficiency and instability. In this paper, we advocate a new direction – *Localized TTS* – that adaptively resamples defective regions while preserving high-quality areas, thereby substantially reducing the search space. This paradigm promises to improve efficiency and stability, but poses two central challenges: *accurately localizing defects* and *maintaining global consistency*. We propose **LoTTS**, the first fully training-free framework for localized TTS. For defect localization, LoTTS detects defective regions by contrasting cross-/self-attention signals under quality-aware prompts (e.g., "high-quality" vs. "low-quality"), reweights them with original prompt attention to filter out irrelevant background, and refines them with self-attention propagation to ensure spatial coherence. For consistency, LoTTS perturbs low-quality regions with noise at intermediate timesteps for localized resampling, and then performs a few global denoising steps to seamlessly couple local corrections with the overall structure and style. Extensive experiments on SD2.1, SDXL, and FLUX demonstrate that LoTTS achieves state-of-the-art performance: it consistently improves both local quality and global fidelity, while reducing GPU cost by 2–4× compared to Best-of-$N$ sampling. These findings establish localized TTS as a promising new direction for scaling diffusion models at inference time.

## 1 Introduction

Diffusion models have become the de-facto standard for high-quality image generation, owing to their strong scalability with data, model size, and compute. This scalability has driven remarkable advances in text-to-image synthesis (Ho et al., 2020; Saharia et al., 2022; Rombach et al., 2022; Ruiz et al., 2023), establishing scaling laws as a guiding principle for building more capable models. While most prior work has focused on scaling at training time, recent studies show that test-time scaling (TTS), allocating additional compute during inference, can also significantly improve sample quality and overall performance (Nichol et al., 2021; Esser et al., 2024; Peebles & Xie, 2023; Ma et al., 2025; Liu et al., 2024b). Despite its promise, existing TTS research remains limited in scope, leaving open questions about how to use inference-time compute more effectively.

Existing TTS methods can be broadly grouped into three categories. The first is *denoising step scaling*, which improves quality by increasing the number of sampling steps (Song et al., 2020b; Lu et al., 2022). However, these gains saturate quickly and plateau around 50 steps, with further increases offering negligible benefit. The second is *Best-of-$N$ search*, where generates $N$ samples and selects the best one by verifier (Liu et al., 2024b; Wang et al., 2023). While simple, this brute-force approach treats each candidate as an independent sample from scratch, overlooking the fact that even imperfect images may be substantially improved through local corrections. As a result, potentially promising samples are discarded, and computation is wasted on redundant global search. The third is *trajectory/noise search*, which perturbs the initial noise or explores alternative sampling

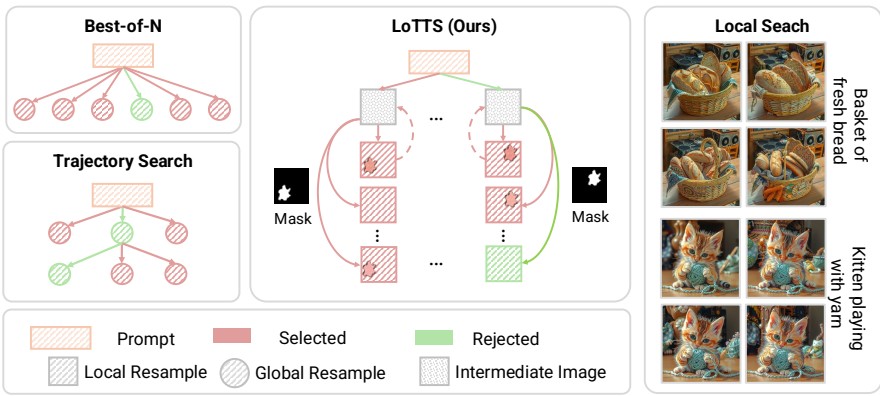

Figure 1: **Overview of the LoTTS framework.** LoTTS utilizes quality-aware masks for localized resampling, contrasting with global Best-of-N and trajectory search. The examples (right) show how LoTTS selectively improves defective regions while preserving high-quality content.

paths (Xu et al., 2023b; Ramesh & Mardani, 2025). Although more fine-grained, it still operates at the full-image level. Searching over all regions can inadvertently disturb areas that are already of high quality, thereby leading to inefficiency and instability. Despite their differences, all three categories share a fundamental limitation: *they operate at the full-image level*, as shown in Figure 1. Consequently, they overlook the inherent spatial heterogeneity of image quality and fail to exploit the potential of localized refinement.

This observation naturally motivates an orthogonal direction: *localized TTS*, where only defective regions are resampled while preserving high-quality content (Cao et al., 2025), thereby substantially reducing the search space. Yet turning this idea into practice introduces two key challenges. The first is *accurate defect localization*: since the distribution of artifacts is complex and prompt-dependent, reliably identifying regions that truly require correction is non-trivial (Liu et al., 2024a; Zhang et al., 2023). The second is *maintaining consistency in local resampling*: as the sampling trajectory is globally defined, locally modifying only a subset of regions may introduce incompatibility, leading to semantic drift, stylistic inconsistency, or boundary artifacts that degrade perceptual quality (Song et al., 2020b; Xu et al., 2023b).

In this paper, we propose **LoTTS** (Prompt-Guided **Lo**calized **T**est-**T**ime **S**caling), the first fully training-free localized TTS framework. Specifically, LoTTS consists of two key components: *defect localization* and *consistency maintenance*. For *defect localization*, LoTTS exploits attention signals from diffusion models to construct a quality-aware mask. We contrast semantically equivalent but quality-differentiated prompts (e.g., "a high-quality image of {p}" vs. "a low-quality image of {p}") and compute the difference between their cross-attention maps, which highlights candidate defective regions (Hertz et al., 2022; Chefer et al., 2023; Chung et al., 2024). To suppress spurious activations, these difference maps are *reweighted with the original prompt attention*, filtering out irrelevant background responses. The resulting signals are then propagated via self-attention to enforce spatial coherence, yielding an automated defect mask without reliance on external predictors or manual annotations. For *consistency maintenance*, LoTTS introduces a localized defect-aware resampling mechanism. After initial sampling, we inject controlled noise into the defective regions of the generated image at an intermediate timestep, followed by localized denoising (Meng et al., 2021). To reconcile local refinements with preserved content, we perform a few global denoising steps, harmonizing style and structure across the image. This ensures that defects are corrected without sacrificing global fidelity. Furthermore, LoTTS is fully plug-and-play and can be seamlessly applied to both diffusion- and flow-based generative models (Song et al., 2020b; Lu et al., 2022).

Our main contributions are summarized as follows:

- We propose LoTTS, the first training-free localized Test-Time Scaling framework, which shifts the focus of inference-time compute from the entire image to defect regions, addressing the inefficiency of global search.

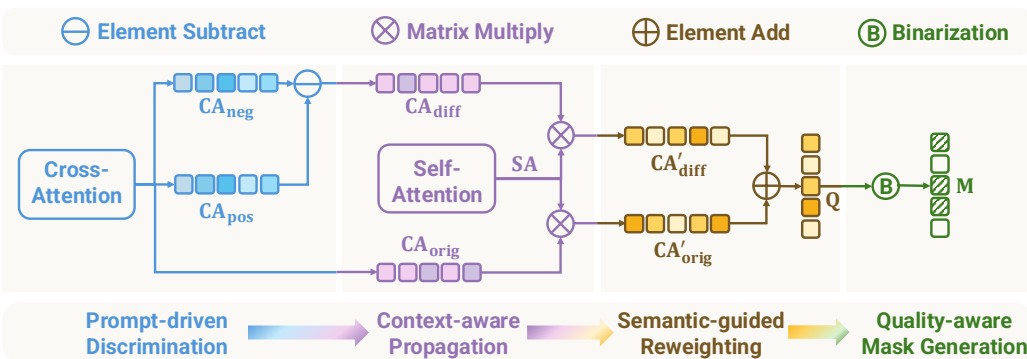

Figure 2: **LoTTS Framework for Defect Localization.** The pipeline consists of four stages: prompt-driven discrimination, context-aware propagation, semantic-guided reweighting, and quality-aware mask generation.

- We introduce an efficient localized noise search mechanism, comprising prompt-driven quality map generation and localized defect-aware resampling.
- We conduct extensive experiments on SD2.1, SDXL, and FLUX, showing that LoTTS consistently achieves state-of-the-art performance across multiple human-preference and automated evaluation metrics, while reducing GPU cost by $2$–$4\times$ relative to Best-of-$N$ sampling.

## 2 RELATED WORK

**Image Generation and Evaluation.** Image generation has advanced from GANs (Goodfellow et al., 2020; Karras et al., 2019) and autoregressive models (van den Oord et al., 2016b;a) to flow-based and diffusion models. GANs achieve high fidelity but are unstable, while autoregressive models capture dependencies yet remain slow. Early flows (Kingma & Dhariwal, 2018) provide exact likelihoods but scale poorly, whereas recent flow matching and rectified flow methods (Lipman et al., 2022; Albergo et al., 2023; Kim et al., 2025a) connect closely to diffusion and offer efficient alternatives. Diffusion models (Ho et al., 2020; Rombach et al., 2022) now dominate, delivering state-of-the-art text-to-image synthesis (Rombach et al., 2022; Podell et al., 2023). In parallel, evaluation methods have evolved: earlier work often collapsed quality into a single global score (Wang et al., 2004; Salimans et al., 2016; Heusel et al., 2017), while spatial indicators relied on texture- or frequency-based heuristics (Yu et al., 2019; Durall et al., 2020) or classifier- and VLM-based predictors (Zhang et al., 2023; Liu et al., 2024a), typically requiring supervision and external datasets. In contrast, LoTTS leverages the diffusion model's inherent attention to localize defects automatically, enabling training-free, localized refinement.

**Generation Quality Enhancement.** Image Editing in modern T2I systems is typically implemented as a post-processing stage. The cascade paradigm, consisting of a generator followed by a refiner, has become a standard design, as in IF (DeepFloyd, 2023), SDXL (Podell et al., 2023) and SD Cascade (Pernias et al., 2023), where an additional pass refines the entire image to enhance global fidelity. However, such refiners usually require separate training and introduce non-negligible computational overhead. In contrast, SDEdit (Meng et al., 2021) shows that diffusion models can perform localized edits via noise–denoise updates within user-specified masks, suggesting the potential of training-free localized refinement. This localized resampling mechanism inspires our approach, which extends SDEdit's manual editing to automated quality-aware refinement at test time.

**Test-Time Scaling in Vision.** Some test-time scaling methods have been proposed to enhance diffusion model generation by allocating more computation at inference. Early work simply increased denoising steps, but improvements saturate quickly beyond a certain number of function evaluations (NFE) (Karras et al., 2022; Song et al., 2020a;b). Recent studies therefore explore alternative directions, such as *Best-of-$N$ search*, where multiple candidates are generated from different noise seeds and a verifier selects the best one (Liu et al., 2024b; Wang et al., 2023), searching over noise initializations (Song et al., 2020b; Xu et al., 2023b), optimizing sampling trajectories (Ramesh &

Mardani, 2025) with verifier feedback (Song et al., 2020a; Karras et al., 2022; Liu et al., 2022; Lu et al., 2022; Salimans & Ho, 2022), or adopting evolutionary (He et al., 2025) and tree-search methods (Yoon et al., 2025). Unlike these approaches, which all operate on the *entire* image and require full regeneration, our LoTTS performs localized TTS by concentrating on low-quality regions for greater efficiency.

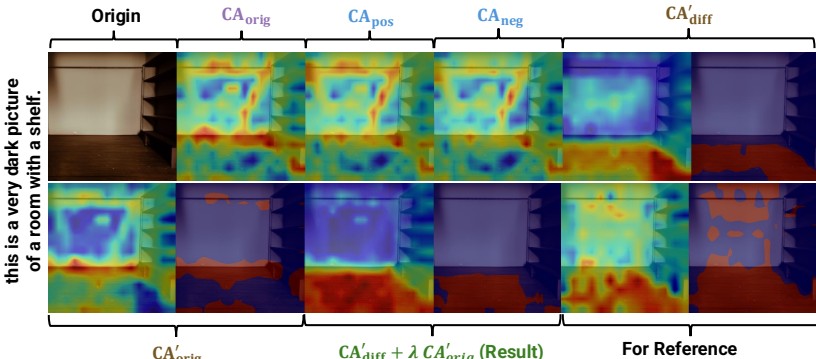

Figure 3: **Visualization of Cross-Attention–based Defect Localization.** From the original ($\text{CA}_{\text{orig}}$), positive ($\text{CA}_{\text{pos}}$), and negative ($\text{CA}_{\text{neg}}$) prompts, LoTTS derives discriminative differences ($\text{CA}'_{\text{diff}}$) and combines them with propagated original attention ($\text{CA}'_{\text{orig}}$) to produce a coherent defect mask ($\text{CA}'_{\text{diff}} + \lambda \text{CA}'_{\text{orig}}$) that highlights low-quality regions. The *Reference* shows a suboptimal supervised baseline (Li et al., 2024).

## 3 PRELIMINARIES

**Diffusion Models.** Diffusion models transform a simple source distribution, *e.g.* a standard Gaussian, into a target data distribution $p_0$. In diffusion models (Sohl-Dickstein et al., 2015; Ho et al., 2020), the forward process gradually corrupts clean data with noise, as

$$\boldsymbol{x}_t = \alpha(t)\boldsymbol{x}_0 + \sigma(t)\epsilon, \quad \epsilon \sim \mathcal{N}(0, I), \tag{1}$$

where $\alpha(t)$ and $\sigma(t)$ denote the noise schedule, and $t \in [0, T]$. To recover data from its diffused representation, diffusion models generally rely on Stochastic Differential Equation(SDE)-based sampling during inference (Song et al., 2020b;a), which introduces stochasticity at every denoising step:

$$\boldsymbol{x}_{t-1} = \sqrt{\alpha(t-1)} \left( \frac{\boldsymbol{x}_t - \sqrt{1-\alpha(t)}\epsilon_\theta(\boldsymbol{x}_t, t)}{\sqrt{\alpha(t)}} \right) + \sqrt{1 - \alpha(t-1) - \sigma(t)^2}\, \epsilon_\theta(\boldsymbol{x}_t, t) + \sigma(t)\epsilon(t). \tag{2}$$

where $\epsilon_\theta(\boldsymbol{x}_t, t)$ denotes the predicted noise at step $t$, $\alpha(t)$ and $\sigma(t)$ are the noise schedule parameters, and $\epsilon_t \sim \mathcal{N}(0, I)$ is a standard Gaussian noise.

**Flow Models.** Flow models (Lipman et al., 2022; Albergo et al., 2023) parameterize the velocity field $u_t \in \mathbb{R}^d$ and generate samples by solving the Flow Ordinary Differential Equation (ODE) (Song et al., 2020b) backward from $t = T$ to $t = 0$:

$$d\boldsymbol{x}_t = u_t(\boldsymbol{x}_t)\, dt. \tag{3}$$

where $u_t(\boldsymbol{x}_t)$ denotes the velocity field learned by the flow model, and $dt$ is an infinitesimal step along the reverse-time ODE. This deterministic dynamics evolves $\boldsymbol{x}_t$ continuously in time, producing identical outcomes for the same input and limiting the applicability of test-time scaling methods (Kim et al., 2025a), which require stochasticity to explore diverse trajectories.

To address this limitation, recent studies propose that the deterministic Flow-ODE could be reformulated into an equivalent SDE (Albergo et al., 2023; Ma et al., 2024; Patel et al., 2024; Kim et al., 2025a; Singh & Fischer, 2024). The resulting stochastic process can be written as:

$$d\boldsymbol{x}_t = \left( u_t(\boldsymbol{x}_t) - \frac{\sigma(t)^2}{2} \nabla \log p_t(\boldsymbol{x}_t) \right) dt + \sigma(t)d\boldsymbol{w}, \tag{4}$$

where the score function $\nabla \log p_t(\boldsymbol{x}_t)$ can be estimated from the velocity field $u_t$ (Singh & Fischer, 2024)), and the Brownian motion term $d\boldsymbol{w}$ introduces stochasticity at each sampling step. This enables LoTTS to naturally apply to Flow-Models.

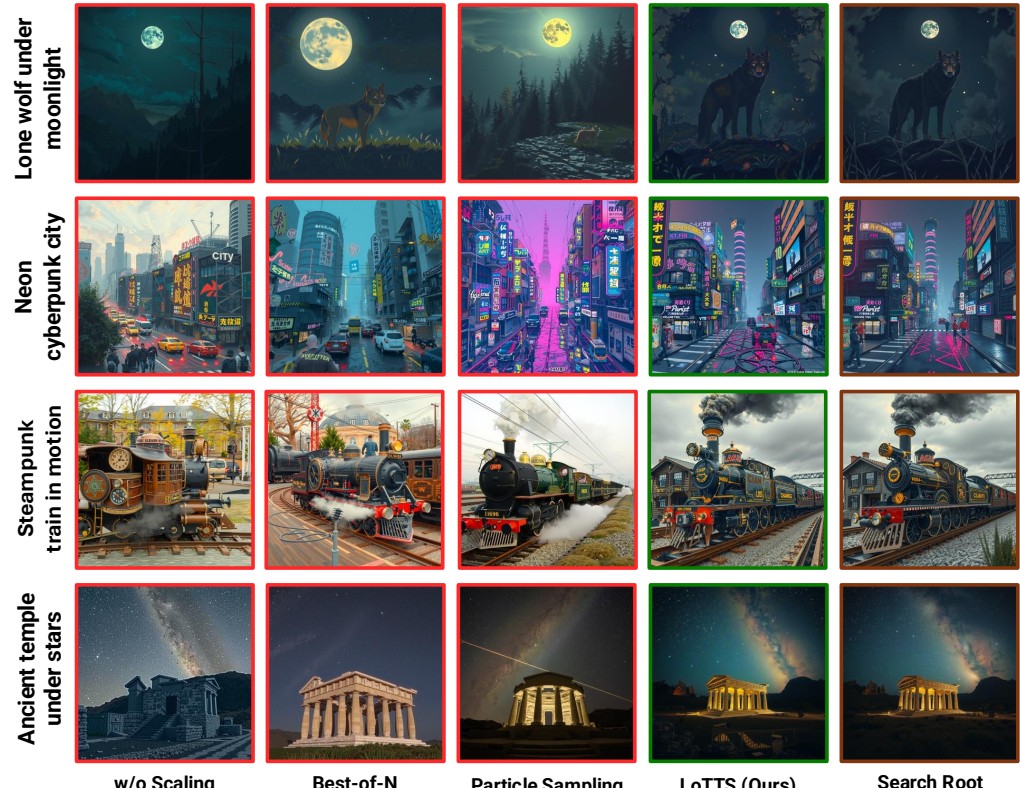

Figure 4: **Qualitative results on challenging text-to-image prompts.** Compared with Resampling, Best-of-$N$ and Particle Sampling, LoTTS better follows complex prompts. Green borders indicate high-quality generations, red mark lower-quality ones, and brown denotes the unrefined baseline (Search Root).

## 4 METHOD

### 4.1 OVERALL

Existing test-time scaling methods, such as Best-of-N or trajectory search, apply uniform resampling to the entire image, overlooking spatial heterogeneity in quality and thereby wasting computation or even degrading well-formed regions. To overcome this, we propose **LoTTS**, a region-aware extension of Best-of-N. Instead of resampling entire images, LoTTS performs localized refinements guided by quality-aware masks, and organizes the search as a hierarchical tree explored via depth-first traversal. The best candidate is then selected as the final output.

To realize this idea, two key challenges must be addressed: (1) *Defect Localization*: how to generate reliable resample masks that identify low-quality regions, and (2) *Consistency Maintenance*: how to perform localized resampling within these regions without disrupting the rest of the image. We describe our solutions to these challenges in the following subsections. For completeness, pseudocode is given in Appendix B, and Appendix G provides a theoretical analysis showing that LoTTS can be proven to achieve a higher expected quality gain under stated conditions.

### 4.2 DEFECT LOCALIZATION

A key challenge in localized test-time scaling is to identify which regions of a generated image truly require refinement. Since image degradations are usually local and diverse, a reliable defect

localization mechanism is indispensable. Existing verifiers typically provide only global quality scores without spatial resolution, making them unsuitable for localized resampling. We leverage the intrinsic attention signals of diffusion models to automatically infer defect regions. The core intuition is that cross-attention with a "low-quality" prompt points to artifact-prone regions, and comparing it with a "high-quality" prompt makes those local defects stand out. To obtain stable and semantically meaningful masks, we further propagate and reweight these signals.

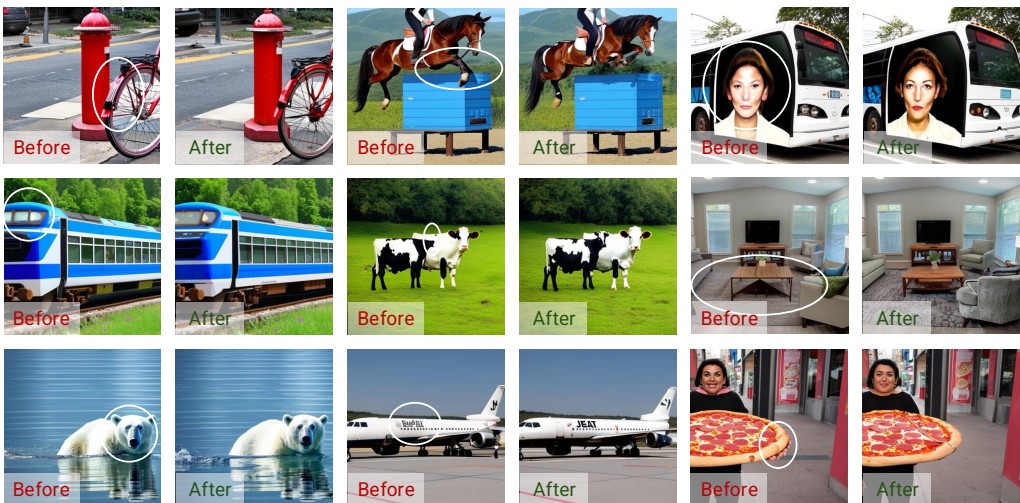

Figure 5: **Before-after comparison in SD2.1.** LoTTS demonstrates strong local refinement capabilities, improving artifacts such as distorted objects, faces, and textures. White circles highlight corrected regions.

**Overview.** As illustrated in Fig. 2, the process has four stages: (1) *Prompt-driven Discrimination*: compute cross-attention maps under high-/low-quality prompts and take their difference to highlight candidate defect regions. (2) *Context-aware Propagation*: refine these signals by propagating across spatially similar positions, mitigating noise, and enforcing local coherence. (3) *Semantic-guided Reweighting*: combine with original prompt attention to suppress irrelevant background. (4) *Quality-aware Mask Generation*: binarize by percentile to control the resampling ratio. Fig. 3 further provides an intuitive visualization of the intermediate results, showing how each step contributes to coherent defect localization.

**Prompt-driven Discrimination.** The intuition is that cross-attention maps reflect how different prompts attend to image regions: a "low-quality" prompt tends to focus on artifact-prone areas, while a "high-quality" prompt attends to cleaner regions. By contrasting the two, we can highlight potential defects.

Following standard practice, we obtain a spatial cross-attention vector $\mathrm{CA} \in \mathbb{R}^S$ by averaging over tokens, heads, and selected layers, where $S = H_s \times W_s$ denotes the number of spatial positions. For a given prompt $p$, we construct three variants: a *positive* prompt ("A high-quality image of $\{p\}$"), a *negative* prompt ("A low-quality image of $\{p\}$"), and the *origin* prompt (the original $p$). This yields three attention vectors $\mathrm{CA}_{\mathrm{pos}}, \mathrm{CA}_{\mathrm{neg}}, \mathrm{CA}_{\mathrm{orig}} \in \mathbb{R}^S$.

We then define a contrastive cross-attention map as

$$\mathrm{CA}_{\mathrm{diff}} = \mathrm{CA}_{\mathrm{neg}} - \mathrm{CA}_{\mathrm{pos}}, \qquad (5)$$

which emphasizes spatial locations where the "bad" prompt receives higher attention than the "good" prompt. The origin vector $\mathrm{CA}_{\mathrm{orig}}$ will later serve as a foreground prior in our aggregation step.

**Context-aware Propagation.** While the contrastive attention map $\mathrm{CA}_{\mathrm{diff}}$ highlights defect-prone regions, it is often noisy and fragmented, with neighboring pixels showing inconsistent scores. Intuitively, spatially or semantically similar regions should share similar quality signals. To enforce such coherence, we propagate the attention scores using a self-attention matrix derived from the image

queries $Q$:

$$\mathrm{CA}' = \mathrm{SA} \times \mathrm{CA}, \qquad \mathrm{SA} = \mathrm{Softmax}\left(\frac{QQ^\top}{\sqrt{d}}\right). \tag{6}$$

This operation smooths the raw attention map by diffusing scores across related spatial positions. Applying it to both $\mathrm{CA}_{\mathrm{diff}}$ and $\mathrm{CA}_{\mathrm{orig}}$ yields refined maps $\mathrm{CA}'_{\mathrm{diff}}$ and $\mathrm{CA}'_{\mathrm{orig}}$ that are more stable and spatially coherent.

**Semantic-guided Reweighting.** Although the contrastive map $\mathrm{CA}'_{\mathrm{diff}}$ can reveal defect-prone areas, it often assigns high scores to background regions with little semantic content (e.g., large sky areas). The intuition is that truly meaningful defects should also lie within the semantic foreground defined by the original prompt. To encode this prior, we combine the contrastive map with the original attention map $\mathrm{CA}'_{\mathrm{orig}}$, which serves as a soft foreground mask:

$$P = \mathrm{CA}'_{\mathrm{diff}} + \lambda \, \mathrm{CA}'_{\mathrm{orig}}, \tag{7}$$

where $\lambda$ balances the defect signal and the foreground prior. The resulting map $P$ emphasizes semantically relevant regions, leading to more reliable defect localization.

**Quality-Aware Mask Generation.** Finally, we need to convert the aggregated quality map into a binary mask that specifies which regions should be resampled. The key idea is to only refine the most degraded areas rather than the entire image. To achieve this, we keep the top $r$ proportion of spatial positions with the highest defect scores, ensuring that resampling is both targeted and controllable. Formally, the mask $M \in \{0, 1\}^S$ is given by

$$M = \mathbb{I}\left(P > \mathrm{Perc}(P, \, 1 - r)\right), \tag{8}$$

where $\mathrm{Perc}(P, \, 1 - r)$ is the $(1 - r)$-quantile of the quality map $P$, and $r \in (0, 1)$ controls the fraction of resampled regions. The mask is then reshaped into the spatial grid $\mathbf{M} \in \{0, 1\}^{H_s \times W_s}$ to guide localized refinement. We compute the attention mask at $t = 0$, matching the final image.

Table 1: **Quantitative results on three benchmarks (Pick-a-Pic, DrawBench, COCO2014).** Across both human-aligned metrics (HPS, AES, Pick, IR) and automated metrics (FID, CLIP), LoTTS consistently outperforms Resampling, Best-of-$N$, and Particle Sampling baselines.

| Model | Method | Pick-a-Pic | | | | DrawBench | | | | COCO2014 | |
|---|---|---|---|---|---|---|---|---|---|---|---|
| | | HPS↑ | AES↑ | Pick↑ | IR↑ | HPS↑ | AES↑ | Pick↑ | IR↑ | FID↓ | CLIP↑ |
| SD2.1 | Resampling | 20.44 | 5.377 | 20.32 | 0.236 | 21.34 | 5.456 | 20.23 | 0.244 | 15.33 | 0.201 |
| | Best-of-$N$ | 21.56 | 5.534 | 21.04 | 0.470 | 22.45 | 5.589 | 20.59 | 0.446 | 13.21 | 0.252 |
| | Particle Sampling | 23.44 | **5.980** | 21.30 | 0.530 | 22.19 | 5.790 | 21.23 | 0.520 | 12.34 | 0.260 |
| | **LoTTS (Ours)** | **24.52** | 5.805 | 21.32 | **0.680** | **23.29** | **5.911** | **21.47** | **0.698** | **10.89** | **0.263** |
| SDXL | Resampling | 23.44 | 6.011 | 21.18 | 0.680 | 23.84 | 6.034 | 21.09 | 0.657 | 9.56 | 0.234 |
| | Best-of-$N$ | 24.54 | 6.198 | 22.01 | 0.790 | 25.27 | 6.238 | 22.23 | 0.756 | 8.34 | 0.268 |
| | Particle Sampling | 25.33 | 6.235 | 22.05 | 0.865 | 26.46 | 6.233 | 22.31 | 0.844 | 7.99 | 0.271 |
| | **LoTTS (Ours)** | **28.23** | **6.304** | **22.30** | **1.102** | **28.90** | **6.321** | **22.38** | **1.111** | **7.33** | **0.297** |
| FLUX | Resampling | 29.34 | 6.298 | 22.07 | 1.038 | 29.28 | 6.223 | 22.05 | 1.100 | 7.01 | 0.282 |
| | Best-of-$N$ | 30.23 | 6.299 | 22.89 | 1.235 | 30.46 | 6.290 | 22.33 | 1.221 | 6.34 | 0.306 |
| | Particle Sampling | 31.56 | **6.532** | 23.31 | 1.450 | 32.28 | 6.523 | 22.90 | 1.445 | 6.02 | 0.332 |
| | **LoTTS (Ours)** | **33.33** | 6.501 | **23.04** | **1.605** | **33.90** | **6.890** | **23.21** | **1.623** | **5.31** | **0.351** |

## 4.3 Consistency Maintenance

With reliable defect masks from the previous step, the next challenge is how to resample the identified regions without disrupting the rest of the image. Restricting updates only to masked regions often introduces boundary artifacts or semantic drift. To address these issues, LoTTS maintains both spatial and temporal consistency during refinement.

**Spatial Consistency.** Resampling in diffusion models typically begins by perturbing the latent representation. If perturbation is applied only within the mask, the noise distribution becomes inconsistent with surrounding regions, creating visible seams. We avoid this by injecting comparable noise into both masked and unmasked areas, which balances noise levels and ensures smooth transitions across boundaries. Formally, given the clean latent $\mathbf{x}_0$ and binary mask $\mathbf{M}$, we initialize the

perturbed latent at timestep $t_0$ as

$$\mathbf{x}_{t_0} = \alpha(t_0)\mathbf{x_0} + \sigma(t_0)((1 - \mathbf{M}) \odot \mathbf{z_{bg}} + \mathbf{M} \odot \mathbf{z_{mask}}) \tag{9}$$

where $\mathbf{z_{bg}} \sim \mathcal{N}(0, I)$ and $\mathbf{z_{mask}} \sim \mathcal{N}(0, I)$.

**Temporal Consistency.** A second challenge is preserving the global semantics of the image. Restarting from pure noise discards structure and forces the model to regenerate the entire scene. Instead, following Meng et al. (2021), we resume denoising from an intermediate step $t_0$, so that global content is retained while localized corrections remain possible. The masked reverse update is

$$\mathbf{x}_{t-\Delta t} = (1 - \mathbf{M}) \odot \big(\alpha(t)\mathbf{x}_0 + \sigma(t)\mathbf{z}\big) + \mathbf{M} \odot \Big(\mathbf{x}_t - \epsilon^2 \epsilon_\theta(\mathbf{x}_t, t) + \epsilon \mathbf{z}\Big). \tag{10}$$

where $\epsilon_\theta(\mathbf{x}_t, t)$ denotes the predicted noise, $\epsilon$ is the per-step noise scale, $N$ is the total number of reverse steps, $\sigma(t)$ is the variance schedule at step $t$, $\mathbf{z} \sim \mathcal{N}(\mathbf{0}, \mathbf{I})$, and $\Delta t = t_0/N$. This keeps unmasked regions faithful while allowing masked regions to be selectively refined.

**Final Integration.** Even after refinement, slight mismatches may remain along boundaries. To ensure seamless blending, we apply a short global denoising sweep. This final pass restores full-image coherence, yielding consistent high-quality outputs.

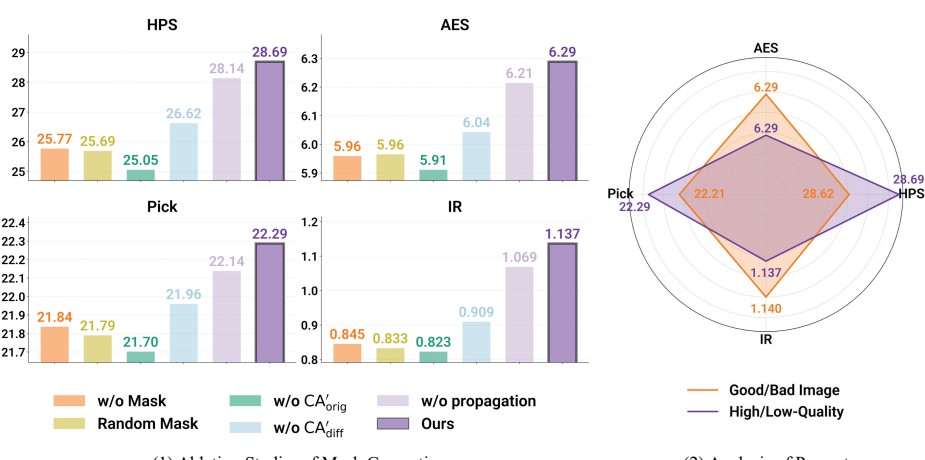

(1) Ablation Studies of Mask Generation    (2) Analysis of Prompt

Figure 6: **Ablation and prompt analysis for LoTTS.** (1) Mask generation ablations show each component contributes to generation performance. (2) Prompt analysis shows comparable results across prompt phrasings, highlighting flexibility.

## 5 EXPERIMENTS

We evaluate LoTTS on three benchmarks: Pick-a-Pic (Kirstain et al., 2023), DrawBench (Saharia et al., 2022), and COCO2014 (Lin et al., 2014). Performance is measured by four human-aligned metrics, HPSv2 (Wu et al., 2023), PickScore (Kirstain et al., 2023), ImageReward (IR) (Xu et al., 2023a) (also serving as the verifier), and Aesthetic Score (AES) (Schuhmann et al., 2022), together with FID (Heusel et al., 2017) and CLIP Score (Radford et al., 2021) on COCO. Experiments are conducted on SD2.1 (Rombach et al., 2022), SDXL (Podell et al., 2023), and Flux.1-schnell (Labs, 2024). SD2.1 and SDXL are diffusion-based, while Flux.1-schnell is a rectified flow-based model. We compare LoTTS with representative sampling and search methods under matched NFE budgets, including vanilla Resampling, Best-of-$N$, and Particle Sampling (Kim et al., 2025b; Singhal et al., 2025). Implementation details are provided in Appendix A.

### 5.1 MAIN RESULTS

Table 1 shows that LoTTS consistently outperforms Resampling, Best-of-$N$, and Particle Sampling (Kim et al., 2025b; Singhal et al., 2025) under matched NFE budgets across SD2.1, SDXL, and FLUX, with clear gains on both human-aligned (HPS, AES, Pick, IR) and automated metrics. Figure 4 further illustrates stronger faithfulness to prompts involving spatial relations, object counts, and fine-grained details. Before–after comparisons are shown in Figure 5 (white circles highlight refined regions). More qualitative results and failure cases provided in Appendices C and F.

## 5.2 ABLATIONS

We ablate two key design choices in LoTTS (Fig. 6). **Mask generation.** Removing any component of the defect localization pipeline (discriminative differences $CA'_{diff}$, propagated original attention $CA'_{orig}$, or context-aware propagation) consistently lowers all four metrics, with the no-mask and random-mask variants yielding the weak performance. **Prompt construction.** Different auxiliary prompts yield similar results, showing LoTTS's robustness to phrasing. Extended ablation results are reported in Appendix D.

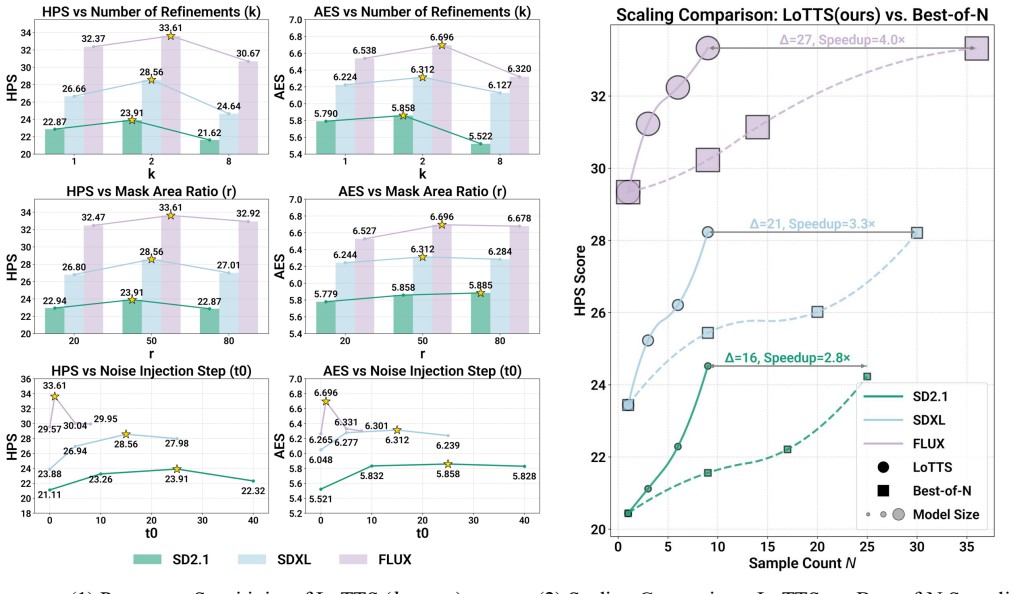

(1) Parameter Sensitivity of LoTTS ($k, r, t_0$)     (2) Scaling Comparison: LoTTS vs. Best-of-N Sampling

Figure 7: **Parameter sensitivity analysis of LoTTS.** Varying localized refinement iterations ($k$), resample area ratio ($r$), and denoising start step ($t_0$) shows that LoTTS maintains stable improvements across HPS, AES, Pick, and IR metrics.

## 5.3 PARAMETER ANALYSIS

We analyze the key hyperparameters of LoTTS: number of refinements ($k$), mask area ratio ($r$), and noise injection step ($t_0$), as shown in Figure 7. Performance peaks at 2 refinements, with additional iterations bringing diminishing returns. Moderate mask ratios around 40–50% achieve the best trade-off, as small ratios miss defects while large ones overwrite clean regions. For noise injection, mid-range steps work best: injecting too early introduces artifacts, while injecting too late weakens refinements. In scaling comparisons, LoTTS achieves similar quality with far fewer samples than Best-of-$N$, matching Best-of-$N$ on SD2.1 and SDXL with $2.8\times$–$3.3\times$ speedups, and reaching up to $4\times$ speedup on FLUX. Extended parameter sensitivity results are reported in Appendix E.

## 6 CONCLUSION

We proposed LoTTS, a training-free framework that extends test-time scaling from global resampling to localized refinement. Unlike conventional methods that apply sampling uniformly, LoTTS leverages defect-aware masks and consistency constraints to focus computation where it matters most. This design not only improves image quality and efficiency but also shows that scaling can be made adaptive to the spatial heterogeneity of generative outputs. Beyond diffusion models, the principle of region-aware scaling provides a general perspective for developing more efficient and controllable inference strategies in generative AI. Our analysis further establishes theoretical conditions under which localized scaling provably outperforms global sampling, and we believe LoTTS opens up promising directions for integrating fine-grained control into test-time algorithms across diverse generative architectures.

ETHICS STATEMENT

All experiments use publicly available datasets and model checkpoints (SD2.1, SDXL, FLUX). No human or animal subjects are involved. The method is intended for research only, and the authors declare no competing interests.

REPRODUCIBILITY STATEMENT

All datasets are public, and implementation details, hyperparameters, and proofs are provided in the appendix. Code will be released to ensure full reproducibility.

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

## A    IMPLEMENTATION DETAILS

We report all settings needed to reproduce our results: hardware/software stack, model checkpoints and preprocessing, default hyperparameters for each evaluation protocol, baselines.

**Hardware and software.**    All experiments were run on NVIDIA A800 (80GB) GPUs; most runs use a single GPU unless otherwise stated.    Mixed precision is enabled via `torch.cuda.amp.autocast` (fp16) unless the metric implementation requires fp32.

**Checkpoints and preprocessing.** We use the official public checkpoints for SD2.1, SDXL, and FLUX.1-schnell. Images are generated at $512{\times}512$ for SD2.1 and at $1024{\times}1024$ for SDXL and FLUX. For latent diffusion backbones, we use the default VAE shipped with each checkpoint. Evaluation images are fed to each metric with the metric's *own* preprocessing. Unless noted, we do not run SDXL-Refiner to avoid confounding the effect of local resampling.

**Default sampler and schedule.** For SD2.1, we use SDE-DPM-Solver++ sampling with 50 NFEs, and CFG scale = 7.5. For SDXL, we use SDE-DPM-Solver++ sampling with 30 NFEs, and CFG scale = 5.5. For FLUX, we use modified SDE-DPM-Solver++ sampling with 10 NFEs, and no CFG. For LoTTS, we inject noise to reach an intermediate step $t_0$ (Sec. 4), then perform local masked refinement followed by a short global integration phase. We keep the original prompt and CFG weight unchanged during refinement.

**Attention hook and mask resolution.** For SD2.1, we hook U-Net cross-attention tensors at the $16{\times}16$ spatial resolution blocks (for $512{\times}512$ images). For SDXL, we hook U-Net cross-attention tensors at the $64{\times}64$ spatial resolution blocks (for $1024{\times}1024$ images). For FLUX, we extract Transformer cross-attention tensors from the $64{\times}64$ resolution of the last 10 blocks (for $1024{\times}1024$ images), and generate attention maps following the method of (Helbling et al., 2025). The Attention maps are averaged across heads, mean-pool across feature dimension, and then average across tokens to obtain a single spatial map per prompt type (pos/neg/origin). Self-attention maps are normalized to $[0, 1]$. The quality map $P$ uses equation 7 in the paper with $\lambda = 0.5$ by default. We compute attention and the final mask at $t{=}0$ (last step), where maps and image latents are already stable. Masks are kept in latent resolution during resampling; only for visualization we bilinearly upsample to pixel space.

**Datasets. Pick-a-Pic:** we follow the official prompt pool for preference-oriented evaluation. **Draw-Bench:** we use all categories and report aggregate metrics. **COCO2014:** we use the widely-used COCO2014-30k.

**Metrics.** We report HPSv2, PickScore, ImageReward, and AES using the authors' official weights and preprocessing. CLIP score and FID are computed with standard settings: CLIP uses ViT-L/14 unless otherwise stated; FID uses Inception-V3 features, 2048-D activations. ImageReward (IR) score is used as the global selection score for Best-of-$N$, LoTTS, and other search-based baselines.

**Baselines and compute budgets.** We compare with Resample, Best-of-N and Particle Sampling under matched NFEs. For "Best-of-$N$" baselines, we sample $N$ candidates *from scratch* and select by the same metric used for reporting to avoid selection bias. For LoTTS we use $k = 2$ localized refinement iterations by default.

**Reproduction checklist.** (1) Use the same checkpoint and VAE; (2) match resolution, sampler, steps, CFG; (3) compute the mask at $t{=}0$ with the same prompt templates; (4) keep $\lambda$, $r$, $t_0$, $T'$, $k$ as in Table 2; (5) fix seeds and determinism flags; (6) run the same metric code and preprocessing.

## B    ALGORITHMIC DETAILS

Algorithm 1 defines how to obtain a binary resample mask $M$ from prompt-conditioned attention maps. Algorithm 2 integrates this mask into localized resampling, yielding improved samples while preserving background. Algorithm 3 further scales LoTTS with DFS-based exploration, iteratively refining candidates and selecting the best according to the global verifier.

| Backbone | Res. | Sampler | Steps | CFG | $\lambda$ | $t_0$ | $T'$ | $k$ | $r(*100)$ |
|---|---|---|---|---|---|---|---|---|---|
| SD 2.1 | 512×512 | SDE-DPM-Solver++ | 50 | 7.5 | 0.50 | 25 | 50 | 2 | 50 |
| SDXL | 1024×1024 | SDE-DPM-Solver++ | 30 | 5.5 | 0.50 | 15 | 30 | 2 | 50 |
| FLUX | 1024×1024 | Modified SDE-DPM-Solver++ | 10 | None | 0.50 | 1 | 10 | 2 | 50 |

Table 2: **Default LoTTS hyperparameters.** *Steps* is the total number of denoising steps in the base sampler; *CFG* is the classifier-free guidance scale; $\lambda$ is the attention contrastive weight; $t_0$ is the noise injection timestep; $T'$ denotes the total refinement steps in LoTTS, defined as local masked denoising plus a small number of global integration steps; $k$ is the number of localized resampling rounds; $r$ is the percentile area ratio used for mask selection.

## B.1 NOTATION

Table 2 summarizes the symbols used throughout the algorithmic description. The goal is to unify notation across Algorithms 1, 2, and 3, so that each variable appearing in the pseudocode has an explicit definition. In particular, we distinguish between user prompts, mask generation parameters, sampling steps, and search-level hyperparameters, as well as intermediate variables such as input/output images and verifier scores.

| Symbol | Description |
|---|---|
| $p$ | original user prompt |
| $p_{\text{pos}}, p_{\text{neg}}$ | quality-aware prompts ("a high-quality image of {$p$}", "a low-quality image of {$p$}") |
| $M$ | binary resample mask (latent resolution) from Algorithm 1 |
| $T$ | total number of diffusion steps |
| $t_0$ | intermediate re-noise step for localized resampling |
| $t_g$ | global integration step |
| $r$ | area ratio used in quantile thresholding |
| $S$ | number of random seeds (global samples at DFS layer 1) |
| $K$ | number of localized refinements per global sample (DFS branching factor) |
| $D$ | DFS maximum depth ($D$=2 in our setting) |
| $N$ | refinement steps within each localized resampling call |
| $x_{\text{in}}, x_{\text{out}}$ | input / output image of LocalizedResample |
| $s_{\text{out}}$ | verifier score of $x_{\text{out}}$ |
| $x_{\text{best}}, v_{\text{best}}$ | current best image and score during DFS search |
| $x^{\star}$ | final best refined image returned by DFS search |
| $\alpha(t), \sigma(t)$ | forward noise schedule parameters |
| $\hat{\epsilon}_{\theta}$ | noise predictor (with classifier-free guidance if enabled) |
| $V(\cdot)$ | global verifier producing scalar score |

Table 3: **Unified notation for Algorithms 1, 2, and 3.**

## B.2 DEFECT LOCALIZATION

Algorithm 1 outlines the **MaskGen** procedure, which constructs a binary mask $M$ indicating defect-prone regions for resampling. The method combines prompt-driven cross-attention discrimination, context-aware propagation, and semantic-guided reweighting to highlight patches where defects are likely to occur. The final mask is obtained by quantile thresholding, ensuring that only a fraction $r$ of the most defective regions are selected. This mask serves as the basis for localized refinement in later algorithms.

---

**Algorithm 1** MaskGen: Defect Localization

---

**Require:** Prompt $p$, positive prompt $p_{\text{pos}}$, negative prompt $p_{\text{neg}}$, selected layers $\mathcal{L}$, foreground weight $\lambda$, area ratio $r$

**Ensure:** Binary mask $M$

1: **Prompt-driven Discrimination:**
2: Compute cross-attention maps: $\text{CA}_{\text{pos}} = \text{CrossAttn}(p_{\text{pos}}, \mathcal{L})$, $\text{CA}_{\text{neg}} = \text{CrossAttn}(p_{\text{neg}}, \mathcal{L})$, $\text{CA}_{\text{orig}} = \text{CrossAttn}(p, \mathcal{L})$
3: $\text{CA}_{\text{diff}} \leftarrow \text{CA}_{\text{neg}} - \text{CA}_{\text{pos}}$
4: **Context-aware Propagation:**
5: $\text{CA}'_{\text{diff}} \leftarrow \text{SelfAttnProp}(\text{CA}_{\text{diff}})$
6: $\text{CA}'_{\text{orig}} \leftarrow \text{SelfAttnProp}(\text{CA}_{\text{orig}})$
7: **Semantic-guided Reweighting:**
8: $P \leftarrow \text{CA}'_{\text{diff}} + \lambda \cdot \text{CA}'_{\text{orig}}$
9: **Mask Generation:**
10: $M \leftarrow \mathbb{I}[P > \text{Perc}(P, 1 - r)]$
11: **return** $M$

---

## B.3 ATTENTION-GUIDED RESAMPLING

Algorithm 2 describes **LocalizedResample**, which integrates the mask $M$ into the diffusion sampling process. Given an input image $x_{\text{in}}$, the method first injects noise to reach an intermediate step $t_0$. During the refinement phase ($t_0 \to t_g$), masked regions are updated using reverse diffusion steps, while unmasked regions are resampled with scheduled Gaussian noise. Finally, a global integration phase ($t_g \to 0$) applies standard reverse diffusion over the entire image to restore coherence. This localized procedure refines defective regions while preserving background content, yielding a new candidate image $x_{\text{out}}$ together with its verifier score $s_{\text{out}}$.

---

**Algorithm 2** LocalizedResample: Attention-Guided Resampling (single refinement)

---

**Require:** Current image $x_{\text{in}}$ (at $t=0$), prompt $p$, total steps $T$, re-noise step $t_0$, global integration step $t_g$, pretrained diffusion model, mask $M$ from Algorithm 1, verifier $V$

**Ensure:** Refined image $x_{\text{out}}$, score $s_{\text{out}}$

1: $x_{\text{anchor}} \leftarrow x_{\text{in}}$
2: $\Delta t \leftarrow t_0/N$          *// masked refinement step size (with $N$ reverse steps)*
3: **Global re-noise to $t_0$**
4: Sample $z_{\text{bg}}, z_{\text{mask}} \sim \mathcal{N}(0, I)$
5: $x_{t_0} \leftarrow \alpha(t_0)\, x_{\text{anchor}} + \sigma(t_0)\, (M \odot z_{\text{mask}} + (1 - M) \odot z_{\text{bg}})$
6: **Masked refinement ($t = t_0 \to t_g$)**
7: **for** $t = t_0, t_0 - \Delta t, \ldots, t_g$ **do**
8:      Sample $z_t \sim \mathcal{N}(0, I)$
9:      $x_{t-\Delta t} \leftarrow (1-M) \odot \big(\alpha(t)\, x_{\text{anchor}} + \sigma(t)\, z_t\big) + M \odot \big(x_t - \Delta t\, \epsilon_\theta(x_t, t; p) + \sigma(t)\, z_t\big)$    *// unmasked: scheduled forward; masked: reverse step*
10: **end for**
11: **Final integration ($t = t_g \to 0$)**
12: **for** $t = t_g, t_g - \Delta t, \ldots, 0$ **do**
13:      Sample $z_t \sim \mathcal{N}(0, I)$
14:      $x_{t-\Delta t} \leftarrow x_t - \Delta t\, \epsilon_\theta(x_t, t; p) + \sigma(t)\, z_t$
15: **end for**
16: $x_{\text{out}} \leftarrow x_0$          *// result after final integration*
17: $s_{\text{out}} \leftarrow V(x_{\text{out}})$
18: **return** $x_{\text{out}}$, $s_{\text{out}}$
*Note: timesteps $t_0 \to 0$ are discretized with step $\Delta t$ and interpolated to match $T$ global steps.*

---

## B.4 OVERALL ALGORITHM OF LoTTS

Algorithm 3 presents the overall **LoTTS** pipeline, formulated as a depth-first search (DFS) with depth $D=2$. At the first level, $S$ global samples are generated from scratch. At the second level,

each global sample undergoes $K$ localized refinements guided by Algorithm 2. During the search, a verifier $V$ evaluates each candidate, and the best image $x^\star$ is maintained and returned at the end. This DFS-based exploration enables systematic search over both global diversity and localized refinement, balancing breadth and depth in test-time scaling.

---

**Algorithm 3** Overall Algorithm of LoTTS

---

**Require:** Prompt $p$, DFS depth $D=2$, number of global seeds $S$, localized branches per seed $K$, total steps $T$, re-noise step $t_0$, global integration step $t_g$, verifier $V$
**Ensure:** Best refined image $x^\star$

1: **function** DFS($p$, $d$, $x$, $x_{\text{best}}$, $v_{\text{best}}$)                                       *// Depth-First-Search*
2:     **if** $x \neq \varnothing$ **then**
3:         $v \leftarrow V(x)$
4:         **if** $v > v_{\text{best}}$ **then**
5:             $x_{\text{best}} \leftarrow x$; $v_{\text{best}} \leftarrow v$
6:         **end if**
7:     **end if**
8:     **if** $d = D$ **then**
9:         **return** ($x_{\text{best}}$, $v_{\text{best}}$)
10:     **end if**
11:     **if** $d = 0$ **then**                                                       *// root $\rightarrow$ global seeds*
12:         **for** $s = 1$ **to** $S$ **do**
13:             $x' \leftarrow$ SampleBase($p$)
14:             ($x_{\text{best}}$, $v_{\text{best}}$) $\leftarrow$ DFS($p$, 1, $x'$, $x_{\text{best}}$, $v_{\text{best}}$)
15:         **end for**
16:         **return** ($x_{\text{best}}$, $v_{\text{best}}$)
17:     **else**                                         *// d=1: global sample $\rightarrow$ K localized refinements*
18:         $M \leftarrow$ MaskGen($x$, $p$)                                             *// Algorithm 1*
19:         **for** $k = 1$ **to** $K$ **do**
20:             ($x'$, $s'$) $\leftarrow$ LocalizedResample($x$, $p$, $T$, $t_0$, $t_g$, $M$, $V$)                  *// Algorithm 2*
21:             ($x_{\text{best}}$, $v_{\text{best}}$) $\leftarrow$ DFS($p$, 2, $x'$, $x_{\text{best}}$, $v_{\text{best}}$)
22:         **end for**
23:         **return** ($x_{\text{best}}$, $v_{\text{best}}$)
24:     **end if**
25: **end function**
26: ($x^\star$, _) $\leftarrow$ DFS($p$, 0, $\varnothing$, $\varnothing$, $-\infty$)
27: **return** $x^\star$

---

## C  EXTENDED QUALITATIVE RESULTS

We provide additional qualitative comparisons to highlight LoTTS's local refinement capabilities.

### C.1  ADDITIONAL MASK GENERATION EXAMPLES

Figures 8 and 9 illustrate the step-by-step mask generation process of LoTTS on SD2.1 and FLUX. Starting from cross-attention maps with the original, positive, and negative prompts, LoTTS computes the discriminative difference ($\text{CA}_{\text{diff}}$), propagates it with self-attention ($\text{CA}'_{\text{orig}}$), and aggregates them to form the final quality-aware mask ($\text{CA}_{\text{diff}} + \text{CA}'_{\text{orig}}$). The results show that LoTTS effectively highlights defective regions while suppressing irrelevant background, producing coherent and semantically aligned masks for localized resampling.

### C.2  BEFORE-AFTER COMPARISON RESULTS

Figures 10 present before-after comparisons in FLUX. LoTTS corrects diverse local artifacts, including distorted body parts, malformed objects, blurred textures, and spurious details, while preserving overall composition and style. White circles mark regions that have been refined.

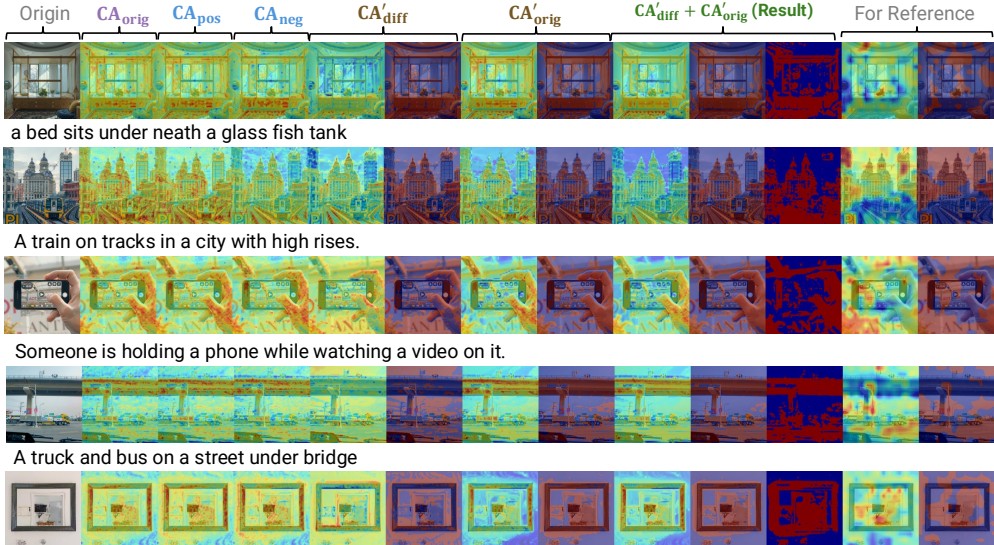

Figure 8: **Additional examples of mask generation in SD2.1.** LoTTS produces reliable quality-aware masks across diverse prompts, effectively localizing low-quality regions for refinement.

Figure 9: **Additional examples of mask generation in FLUX.** LoTTS produces finer and more detailed masks in FLUX, consistently localizing local degradations under varied prompts and enabling more precise localized resampling.

# D  ABLATIONS

We conduct ablation studies to evaluate the contribution of different components in our mask generation pipeline. Table 4 reports results on Pick-a-Pic and DrawBench across SD2.1, SDXL, and FLUX. Removing or altering components (e.g., w/o Mask, Random Mask, w/o $CA'_{orig}$, w/o $CA_{diff}$, or w/o Propagation) consistently degrades performance, confirming the effectiveness of our full design. Figure 11 further visualizes these results, showing that LoTTS achieves the best performance across all metrics, highlighting the importance of each mask generation step.

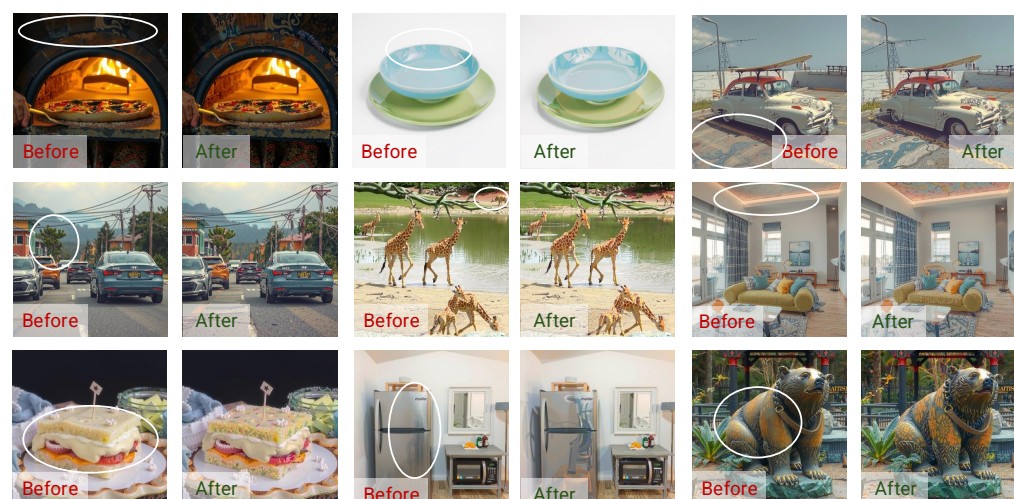

Figure 10: **Before-after comparison in FLUX.** LoTTS achieves consistent local refinements across diverse prompts, enhancing structural integrity, semantics, and perceptual quality. White circles highlight corrected regions.

Table 4: **Ablation study of mask generation strategies on Pick-a-Pic and DrawBench.** Results for SD2.1, SDXL, and FLUX show that removing or altering components (e.g., w/o Mask, Random Mask, w/o $\text{CA}'_{\text{orig}}$, w/o $\text{CA}_{\text{diff}}$, or w/o Propagation) degrades performance, confirming the effectiveness of our full design.

| Model | Mask Strategy | Pick-a-Pic | | | | DrawBench | | | |
|---|---|---|---|---|---|---|---|---|---|
| | | HPS↑ | AES↑ | Pick↑ | IR↑ | HPS↑ | AES↑ | Pick↑ | IR↑ |
| SD2.1 | w/o Mask | 21.21 | 5.245 | 21.14 | 0.452 | 22.33 | 5.591 | 20.66 | 0.453 |
| | Random Mask | 20.59 | 5.325 | 21.01 | 0.457 | 22.34 | 5.580 | 20.67 | 0.452 |
| | w/o $\text{CA}'_{\text{orig}}$ | 20.43 | 5.223 | 20.87 | 0.444 | 21.22 | 5.570 | 20.44 | 0.445 |
| | w/o $\text{CA}'_{\text{diff}}$ | 21.31 | 5.303 | 20.95 | 0.458 | 22.53 | 5.605 | 21.01 | 0.500 |
| | w/o Propagation | 23.51 | 5.705 | 21.12 | 0.600 | 22.89 | 5.811 | 21.36 | 0.688 |
| | **Ours** | **24.52** | **5.805** | **21.32** | **0.680** | **23.29** | **5.911** | **21.47** | **0.698** |
| SDXL | w/o Mask | 24.20 | 6.120 | 22.04 | 0.780 | 25.31 | 6.244 | 22.31 | 0.788 |
| | Random Mask | 24.19 | 6.115 | 22.01 | 0.755 | 25.65 | 6.231 | 22.32 | 0.778 |
| | w/o $\text{CA}'_{\text{orig}}$ | 24.00 | 6.100 | 22.00 | 0.748 | 24.55 | 6.222 | 22.28 | 0.766 |
| | w/o $\text{CA}'_{\text{diff}}$ | 26.00 | 6.201 | 22.05 | 0.850 | 26.35 | 6.272 | 22.30 | 0.901 |
| | w/o Propagation | 28.11 | 6.290 | 22.11 | 1.001 | 28.30 | 6.301 | 22.35 | 1.100 |
| | **Ours** | **28.23** | **6.304** | **22.30** | **1.102** | **28.90** | **6.321** | **22.38** | **1.111** |
| FLUX | w/o Mask | 30.25 | 6.324 | 22.67 | 1.241 | 31.32 | 6.231 | 22.20 | 1.357 |
| | Random Mask | 30.15 | 6.313 | 22.55 | 1.211 | 31.23 | 6.225 | 22.18 | 1.344 |
| | w/o $\text{CA}'_{\text{orig}}$ | 30.00 | 6.131 | 22.45 | 1.200 | 30.13 | 6.220 | 22.17 | 1.333 |
| | w/o $\text{CA}'_{\text{diff}}$ | 31.55 | 6.345 | 22.64 | 1.315 | 32.00 | 6.530 | 22.81 | 1.433 |
| | w/o Propagation | 33.23 | 6.399 | 22.88 | 1.501 | 32.80 | 6.780 | 23.01 | 1.523 |
| | **Ours** | **33.33** | **6.501** | **23.04** | **1.605** | **33.90** | **6.890** | **23.21** | **1.623** |

# E PARAMETER ANALYSIS

We analyze the effect of LoTTS hyperparameters and compare its scaling behavior with Best-of-$N$. Tables 5–7, Tables 8–9, and Figure 12 summarize the results.

**Iteration number $k$.** Performance consistently peaks at $k=2$ across SD2.1, SDXL, and FLUX. Increasing to $k=8$ leads to noticeable drops, showing that only a small number of localized refinements is effective.

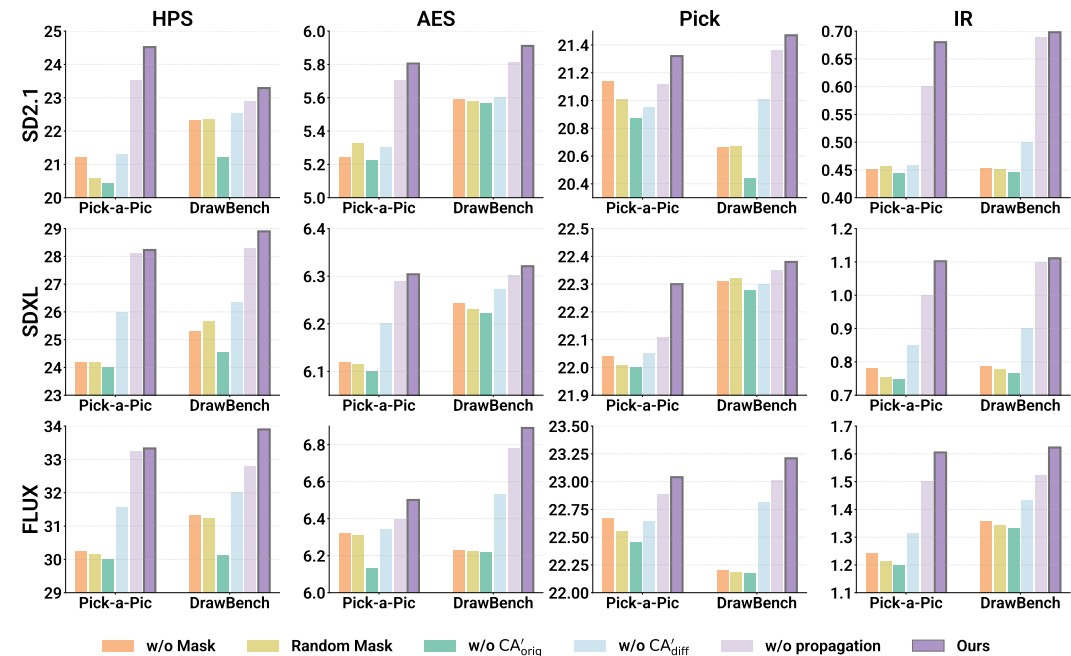

Figure 11: **Visualization of mask strategy ablations.** Quantitative results on Pick-a-Pic and Draw-Bench across SD2.1, SDXL, and FLUX for HPS, AES, Pick, and IR metrics. Compared variants include w/o Mask, Random Mask, w/o $\text{CA}'_{\text{orig}}$, w/o $\text{CA}_{\text{diff}}$, and w/o Propagation. LoTTS consistently outperforms these ablated versions, highlighting the importance of each mask generation component.

**Mask area ratio** $r$**.** Moderate ratios around 40–50% give the best trade-off. Too small ratios miss defects, while too large ratios overwrite clean regions and slightly reduce efficiency.

**Noise injection step** $t_0$**.** Intermediate values yield the strongest results: very early injection may introduce artifacts, while very late injection weakens refinement. The best-performing points appear around $t_0=25$ for SD2.1 (out of 50 steps), $t_0=15$ for SDXL (out of 30 steps), and $t_0=1$ for FLUX (out of 10 steps), corresponding roughly to the early-to-mid stages of the denoising process.

**Visualization.** Figure 12 illustrates these trends, with best configurations marked by stars.

**Scaling comparison.** Table 8 shows that LoTTS scales efficiently with sample count $S$, achieving strong improvements even with small $S$. In contrast, Table 9 indicates that Best-of-$N$ needs much larger $N$ to reach similar quality. For example, LoTTS with $S=9$ matches or surpasses Best-of-25 on SD2.1, Best-of-30 on SDXL, and Best-of-36 on FLUX, while requiring 3–4× fewer samples.

Overall, these results demonstrate that LoTTS achieves robust gains across datasets and consistently outperforms global Best-of-$N$ in both effectiveness and efficiency.

## F  FAILURE CASES AND DIAGNOSTICS

While LoTTS corrects many local artifacts, several failure modes remain. As shown in Figure 13, complex geometry near boundaries can remain distorted (e.g., the cat face). In scenes requiring global coherence or in low-texture areas, refinements may introduce or amplify artifacts, such as banding in sky or stains on the tennis court floor. In addition, improvements are limited when defects are subtle or missed by the mask, as in the Big Ben example where before and after differ little. These issues, though relatively rare, highlight the need for stronger global consistency, artifact suppression, and higher-recall masks.

Table 5: **Quantitative results w.r.t. number of refinements** $k$. Results on Pick-a-Pic and Draw-Bench for SD2.1, SDXL, and FLUX.

| Model | $k$ | Pick-a-Pic | | | | DrawBench | | | |
|---|---|---|---|---|---|---|---|---|---|
| | | HPS↑ | AES↑ | Pick↑ | IR↑ | HPS↑ | AES↑ | Pick↑ | IR↑ |
| SD2.1 | 1 | 23.50 | 5.781 | 21.20 | 0.581 | 22.23 | 5.799 | 21.24 | 0.540 |
| | 2 | **24.52** | **5.805** | **21.32** | **0.680** | **23.29** | **5.911** | **21.47** | **0.698** |
| | 8 | 21.34 | 5.367 | 20.43 | 0.363 | 21.90 | 5.678 | 20.38 | 0.354 |
| SDXL | 1 | 26.31 | 6.205 | 22.15 | 0.965 | 27.01 | 6.243 | 22.21 | 0.944 |
| | 2 | **28.23** | **6.304** | **22.30** | **1.102** | **28.90** | **6.321** | **22.38** | **1.111** |
| | 8 | 24.45 | 6.111 | 21.53 | 0.781 | 24.84 | 6.144 | 21.56 | 0.877 |
| FLUX | 1 | 31.46 | 6.445 | 22.89 | 1.550 | 33.28 | 6.631 | 22.95 | 1.457 |
| | 2 | **33.33** | **6.501** | **23.04** | **1.605** | **33.90** | **6.890** | **23.21** | **1.623** |
| | 8 | 30.54 | 6.308 | 22.57 | 1.244 | 30.81 | 6.332 | 22.15 | 1.200 |

Table 6: **Quantitative results w.r.t. mask area ratio** $r$. Results on Pick-a-Pic and DrawBench for SD2.1, SDXL, and FLUX.

| Model | $r$ | Pick-a-Pic | | | | DrawBench | | | |
|---|---|---|---|---|---|---|---|---|---|
| | | HPS↑ | AES↑ | Pick↑ | IR↑ | HPS↑ | AES↑ | Pick↑ | IR↑ |
| SD2.1 | 20 | 23.29 | 5.780 | 21.22 | 0.630 | 22.59 | 5.778 | 21.33 | 0.532 |
| | 50 | **24.52** | 5.805 | 21.32 | **0.680** | **23.29** | **5.911** | **21.47** | **0.698** |
| | 80 | 23.34 | **5.978** | 21.23 | 0.640 | 22.39 | 5.791 | 21.43 | 0.620 |
| SDXL | 20 | 26.13 | 6.244 | 22.15 | 0.895 | 27.46 | 6.243 | 22.13 | 0.894 |
| | 50 | **28.23** | **6.304** | **22.30** | **1.102** | **28.90** | **6.321** | **22.38** | **1.111** |
| | 80 | 27.03 | 6.256 | 22.13 | 0.985 | 26.98 | 6.313 | 22.21 | 0.944 |
| FLUX | 20 | 31.46 | 6.432 | 22.99 | 1.550 | 33.48 | 6.623 | 22.89 | 1.545 |
| | 50 | **33.33** | **6.501** | **23.04** | **1.605** | **33.90** | **6.890** | **23.21** | **1.623** |
| | 80 | 32.66 | **6.632** | 23.00 | 1.459 | 33.18 | 6.724 | 22.93 | 1.615 |

# G THEORETICAL ANALYSIS: WHEN LOCALIZED TTS OUTPERFORMS GLOBAL TTS

## G.1 NOTATION

For clarity, Table 10 summarizes the variables used throughout this section. This ensures every symbol in the derivation has an explicit meaning.

## G.2 QUALITY DECOMPOSITION

Before comparing global and local strategies, we first relate global quality $r(x)$ to patch-level scores. To lower bound global improvement by local improvements, we adopt a standard additive inequality:

$$r(x) \geq \sum_{j=1}^{M} w_j\, r_j(x_j), \tag{11}$$

where $w_j \geq 0$ are weights reflecting importance of each patch. This allows us to measure global gain through weighted patch-level gains.

Table 7: **Quantitative results w.r.t. noise injection step** $t_0$**.** Results on Pick-a-Pic and DrawBench for SD2.1, SDXL, and FLUX.

| Model | $t_0$ | Pick-a-Pic | | | | DrawBench | | | |
|---|---|---|---|---|---|---|---|---|---|
| | | HPS↑ | AES↑ | Pick↑ | IR↑ | HPS↑ | AES↑ | Pick↑ | IR↑ |
| SD2.1 | 0 | 20.54 | 5.487 | 20.53 | 0.238 | 21.68 | 5.556 | 20.33 | 0.254 |
| | 10 | 23.32 | 5.755 | 21.22 | 0.678 | 23.19 | 5.910 | 21.07 | 0.658 |
| | 25 | **24.52** | **5.805** | **21.32** | **0.680** | **23.29** | **5.911** | **21.47** | **0.698** |
| | 40 | 21.42 | 5.775 | 21.01 | 0.650 | 23.21 | 5.881 | 21.27 | 0.648 |
| SDXL | 0 | 23.81 | 6.041 | 21.34 | 0.683 | 23.95 | 6.054 | 21.14 | 0.667 |
| | 5 | 25.13 | 6.254 | 22.17 | 1.062 | 28.75 | 6.301 | 21.08 | 1.051 |
| | 15 | **28.23** | **6.304** | **22.30** | **1.102** | **28.90** | **6.321** | **22.38** | **1.111** |
| | 25 | 27.30 | 6.194 | 22.24 | 1.002 | 28.65 | 6.283 | 21.88 | 1.091 |
| FLUX | 0 | 29.55 | 6.298 | 22.57 | 1.111 | 29.59 | 6.233 | 22.15 | 1.117 |
| | 1 | **33.33** | **6.501** | **23.04** | **1.605** | **33.90** | **6.890** | **23.21** | **1.623** |
| | 5 | 30.15 | 6.312 | 22.77 | 1.211 | 29.93 | 6.349 | 22.48 | 1.263 |
| | 8 | 30.01 | 6.301 | 22.60 | 1.200 | 29.88 | 6.301 | 22.30 | 1.145 |

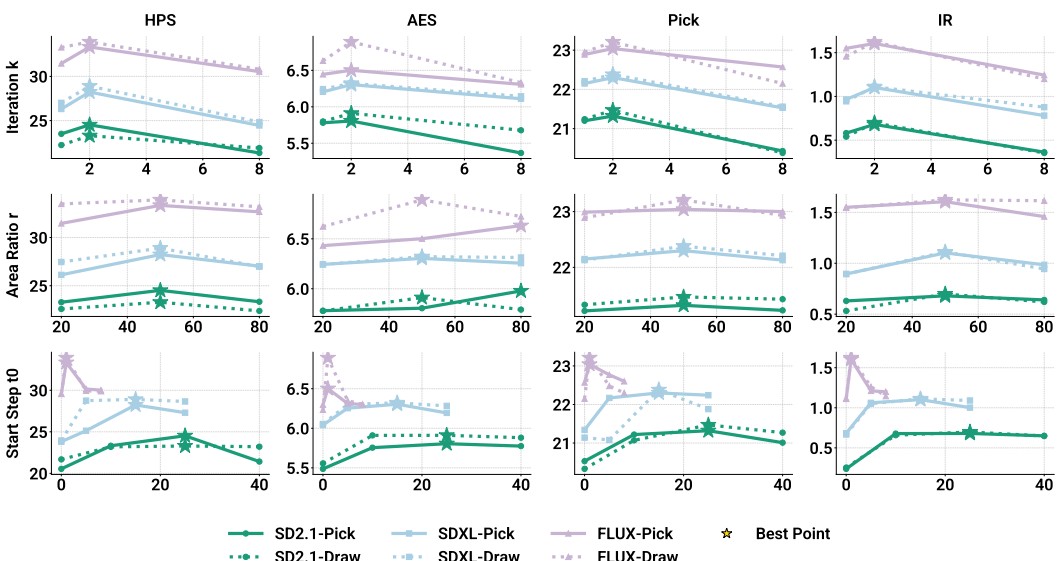

Figure 12: **Hyperparameter analysis of LoTTS.** Performance on Pick-a-Pic and DrawBench with respect to (top) the number of refinements $k$, (middle) mask area ratio $r$, and (bottom) noise injection step $t_0$, across SD2.1, SDXL, and FLUX. Stars mark the best-performing configurations.

### G.3 FROM PRECISION/RECALL TO SELECTION SIZE

We next connect the mask's precision/recall to expected true/false selections. This step is necessary to compute how many patches are truly improved versus unnecessarily modified.

**Lemma 1** (Expected TP/FP size)**.** *Let* $S \subseteq [M]$ *be the (fixed) set of truly defective patches with* $|S| = s$*, and let* $\widehat{S}$ *be the random set selected by the mask. Define*

$$\mathrm{TP} := |S \cap \widehat{S}|, \qquad \mathrm{FP} := |\widehat{S} \setminus S| = |\widehat{S}| - \mathrm{TP}.$$

*Assume recall* $\rho := \frac{\mathbb{E}[\mathrm{TP}]}{s} \in [0,1]$ *and (operational) precision* $\pi := \frac{\mathbb{E}[\mathrm{TP}]}{\mathbb{E}[|\widehat{S}|]} \in (0,1]$*. Then*

$$\mathbb{E}[\mathrm{TP}] = \rho s, \qquad \mathbb{E}[|\widehat{S}|] = \frac{\rho s}{\pi}, \qquad \mathbb{E}[\mathrm{FP}] = \rho s \left( \frac{1}{\pi} - 1 \right).$$

Table 8: **Quantitative results of LoTTS w.r.t. sample count $S$.** Results on Pick-a-Pic and Draw-Bench for SD2.1, SDXL, and FLUX.

| Model | $S$ | Pick-a-Pic | | | | DrawBench | | | |
|---|---|---|---|---|---|---|---|---|---|
| | | HPS↑ | AES↑ | Pick↑ | IR↑ | HPS↑ | AES↑ | Pick↑ | IR↑ |
| SD2.1 | 1 | 20.44 | 5.377 | 20.32 | 0.236 | 21.34 | 4.546 | 20.23 | 0.244 |
| | 3 | 21.12 | 5.532 | 20.56 | 0.413 | 21.88 | 5.623 | 20.54 | 0.398 |
| | 6 | 22.29 | 5.712 | 20.99 | 0.531 | 21.45 | 5.807 | 21.00 | 0.591 |
| | 9 | **24.52** | **5.805** | **21.32** | **0.680** | **23.29** | **5.911** | **21.47** | **0.698** |
| SDXL | 1 | 23.44 | 6.011 | 21.18 | 0.680 | 23.84 | 6.034 | 21.09 | 0.657 |
| | 3 | 25.23 | 6.124 | 21.55 | 0.712 | 25.11 | 6.123 | 21.55 | 0.701 |
| | 6 | 26.21 | 6.153 | 21.89 | 0.813 | 26.23 | 6.241 | 21.98 | 0.812 |
| | 9 | **28.23** | **6.304** | **22.30** | **1.102** | **28.90** | **6.321** | **22.38** | **1.111** |
| FLUX | 1 | 29.34 | 6.298 | 22.07 | 1.038 | 29.28 | 6.223 | 22.05 | 1.100 |
| | 3 | 31.23 | 6.370 | 22.53 | 1.203 | 30.45 | 6.350 | 22.47 | 1.321 |
| | 6 | 32.24 | 6.434 | 22.88 | 1.523 | 32.23 | 6.591 | 23.01 | 1.521 |
| | 9 | **33.33** | **6.501** | **23.04** | **1.605** | **33.90** | **6.890** | **23.21** | **1.623** |

Table 9: **Quantitative results of Best-of-$N$ baseline w.r.t. sample count $S$.** Results on Pick-a-Pic and DrawBench for SD2.1, SDXL, and FLUX.

| Model | $S$ | Pick-a-Pic | | | | DrawBench | | | |
|---|---|---|---|---|---|---|---|---|---|
| | | HPS↑ | AES↑ | Pick↑ | IR↑ | HPS↑ | AES↑ | Pick↑ | IR↑ |
| SD2.1 | 1 | 20.44 | 5.377 | 20.32 | 0.236 | 21.34 | 4.546 | 20.23 | 0.244 |
| | 9 | 21.56 | 5.534 | 21.04 | 0.470 | 22.45 | 5.589 | 20.59 | 0.446 |
| | 17 | 22.21 | 5.755 | 21.11 | 0.524 | 25.17 | 5.817 | 21.10 | 0.602 |
| | 25 | **24.23** | **5.825** | **21.26** | **0.681** | **23.29** | **5.911** | **21.47** | **0.701** |
| SDXL | 1 | 23.44 | 6.011 | 21.18 | 0.680 | 23.84 | 6.034 | 21.09 | 0.657 |
| | 9 | 25.44 | 6.198 | 22.01 | 0.790 | 25.27 | 6.238 | 22.03 | 0.756 |
| | 20 | 26.02 | 6.201 | 22.11 | 0.833 | 26.31 | 6.233 | 22.30 | 0.832 |
| | 30 | **28.21** | **6.304** | **22.21** | **1.112** | **28.90** | **6.321** | **22.38** | **1.113** |
| FLUX | 1 | 29.34 | 6.298 | 22.07 | 1.038 | 29.28 | 6.223 | 22.05 | 1.100 |
| | 9 | 30.23 | 6.299 | 22.89 | 1.235 | 30.46 | 6.290 | 22.23 | 1.221 |
| | 14 | 31.14 | 6.343 | 22.93 | 1.521 | 32.90 | 6.599 | 22.98 | 1.533 |
| | 36 | **33.32** | **6.531** | **23.13** | **1.615** | **33.90** | **6.890** | **23.21** | **1.620** |

*Proof.* By the definition of recall $\rho = \frac{\mathbb{E}[\text{TP}]}{s}$, we immediately have

$$\mathbb{E}[\text{TP}] = \rho\,s.$$

By the (operational) definition of precision $\pi = \frac{\mathbb{E}[\text{TP}]}{\mathbb{E}[|\widehat{S}|]}$, we get

$$\mathbb{E}[|\widehat{S}|] = \frac{\mathbb{E}[\text{TP}]}{\pi} = \frac{\rho\,s}{\pi}.$$

Finally, using linearity of expectation and $\text{FP} = |\widehat{S}| - \text{TP}$,

$$\mathbb{E}[\text{FP}] = \mathbb{E}[|\widehat{S}|] - \mathbb{E}[\text{TP}] = \frac{\rho\,s}{\pi} - \rho\,s = \rho\,s\left(\frac{1}{\pi} - 1\right).$$

$\square$

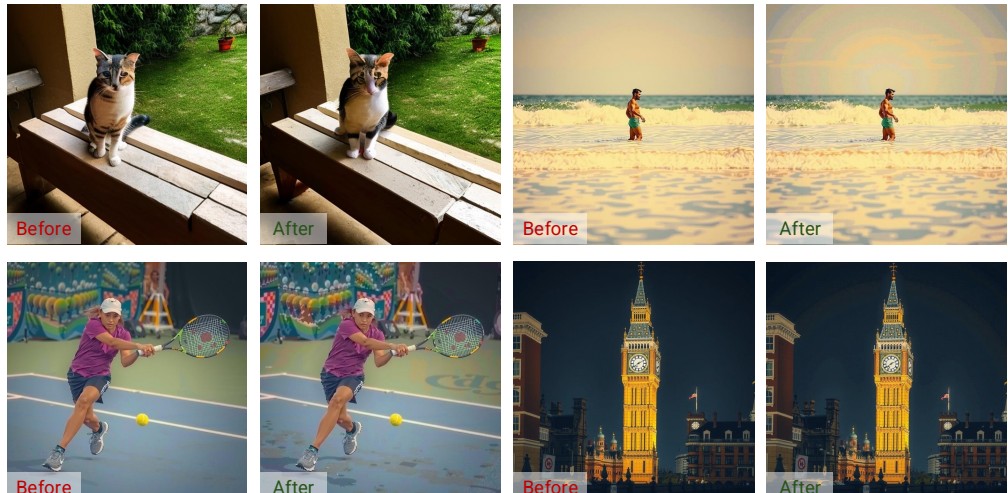

Figure 13: **Failure cases of LoTTS.** While LoTTS reduces many local artifacts, a few failures remain in boundary geometry, in scenes requiring global coherence or containing low-texture regions (e.g., sky, court floor), and in cases with only subtle defects where refinements bring limited improvements.

| Symbol | Description |
|---|---|
| $M$ | total number of non-overlapping image patches |
| $s$ | number of defective patches ($s \ll M$) |
| $S$ | set of defective patches, with $|S| = s$ |
| $\widehat{S}$ | set of patches selected by LoTTS for resampling |
| $\pi$ | precision: fraction of selected patches that are truly defective |
| $\rho$ | recall: fraction of defective patches successfully selected |
| $x$ | generated image; $x_j$ denotes patch $j$ |
| $r(x)$ | perceptual quality functional of image $x$ |
| $r_j(x_j)$ | patch-level quality score |
| $w_j$ | importance weight of patch $j$ in quality lower bound |
| $\delta_j$ | expected gain if defective patch $j$ is repaired |
| $\gamma_j$ | expected loss if clean patch $j$ is harmed |
| $\overline{\delta}$ | average weighted repair gain over defective patches |
| $\overline{\gamma}$ | average weighted harm over clean patches |
| $\theta_g$ | probability of repairing a defective patch under global resampling |
| $q$ | probability of repairing a defective patch under localized resampling |
| $h_g$ | probability of harming a clean patch under global resampling |
| $h_\ell$ | probability of harming a clean patch under localized resampling |
| $C_g$ | compute cost of one global resampling trial |
| $C_\ell$ | compute cost of one localized resampling trial |
| $B$ | total compute budget (e.g., measured in NFEs) |

Table 10: **Notation used in the theoretical analysis.**

**Discussion.**

- **Why use $\pi = \frac{\mathbb{E}[\text{TP}]}{\mathbb{E}[|\widehat{S}|]}$?** Precision is often defined as $\mathbb{E}\left[\frac{\text{TP}}{|\widehat{S}|}\mathbf{1}\{|\widehat{S}| > 0\}\right]$. To avoid division-by-zero and keep algebra tractable, we adopt the *ratio-of-expectations* form, which is standard in compute-budget analyses. If one insists on the expectation-of-ratio definition, Jensen-type arguments yield the bound $\mathbb{E}[|\widehat{S}|] \geq \frac{\mathbb{E}[\text{TP}]}{\pi}$, so our equalities become tight upper/lower bounds; the conclusions below only change by harmless inequalities.

- **Edge cases.** $\pi = 1$ implies no false positives, hence $\mathbb{E}[\text{FP}] = 0$. $\rho = 1$ means all $s$ defects are captured in expectation, i.e., $\mathbb{E}[\text{TP}] = s$. We exclude the degenerate $\pi = 0$ case since precision 0 implies $\mathbb{E}[\text{TP}] = 0$ or $\mathbb{E}[|\widehat{S}|] = \infty$.

- **What randomness is averaged over?** Expectations are taken over the randomness of mask construction (and, if applicable, sampling noise). No independence assumptions are required; linearity of expectation suffices.

### G.4 PER-TRIAL EXPECTED GAIN

We now compute the expected gain of a single trial, either by global resampling or localized resampling. This quantifies the balance between repairing defective patches and potentially harming clean ones.

Let $x$ denote the image before a trial and $x^+$ the image after the trial. Define the gain as

$$\Delta r := r(x^+) - r(x).$$

Using the local decomposability lower bound

$$r(x) \geq \sum_{j=1}^{M} w_j \, r_j(x_j),$$

we obtain

$$\Delta r \geq \sum_{j=1}^{M} w_j \big( r_j(x_j^+) - r_j(x_j) \big).$$

For each defective patch $j \in S$: - if repaired, the expected improvement is $\delta_j \geq 0$. For each clean patch $j \notin S$: - if harmed, the expected loss is $\gamma_j \geq 0$.

Define the weighted averages

$$\overline{\delta} := \frac{1}{s} \sum_{j \in S} w_j \delta_j, \qquad \overline{\gamma} := \frac{1}{M-s} \sum_{j \notin S} w_j \gamma_j,$$

where $s := |S| \ll M$ is the number of defective patches.

**Global resampling.** In global test-time scaling, all patches are resampled: - each defective patch $j \in S$ is repaired with probability $\theta_g$, - each clean patch $j \notin S$ is harmed with probability $h_g$.

The expected gain is then lower bounded by

$$\mathbb{E}[\Delta r]_{\text{global}} \geq s \, \theta_g \, \overline{\delta} - (M-s) \, h_g \, \overline{\gamma}. \tag{12}$$

Here the positive term reflects that all $s$ defects may be repaired, while the negative term reflects that all $(M-s)$ clean patches are simultaneously at risk.

**Localized resampling (LoTTS).** In localized test-time scaling, only patches selected by the mask $\widehat{S}$ are resampled: - the *true positives* $S \cap \widehat{S}$ may be repaired, - the *false positives* $\widehat{S} \setminus S$ may be harmed.

Let $\rho$ denote recall, i.e., $\rho = \frac{\mathbb{E}[|S \cap \widehat{S}|]}{s}$, and $\pi$ denote precision, i.e., $\pi = \frac{\mathbb{E}[|S \cap \widehat{S}|]}{\mathbb{E}[|\widehat{S}|]}$. Let $q$ be the probability of repairing a defective patch under localized resampling, and $h_\ell$ the probability of harming a selected clean patch.

By linearity of expectation, and using Lemma 1, the expected number of true positives is $\rho s$, while the expected number of false positives is $\rho s(\frac{1}{\pi} - 1)$. Hence,

$$\mathbb{E}[\Delta r]_{\text{local}} \geq \rho \, s \, q \, \overline{\delta} - \rho \, s \left( \frac{1}{\pi} - 1 \right) h_\ell \, \overline{\gamma}. \tag{13}$$

**Interpretation.** The global method has the potential to repair all $s$ defective patches, but it also incurs a harm penalty that scales with the large number of clean patches $(M-s)$. In contrast, LoTTS can only repair $\rho s$ defects on average (recall-weighted), yet its harm penalty grows only with the expected number of false positives, $\rho s(\frac{1}{\pi} - 1)$, which scales with $s$ rather than $M$. This asymmetry highlights the *sparse-defect advantage*: when $s \ll M$, localized resampling avoids the heavy global penalty on many clean regions, focusing compute where it matters most.

## G.5 BUDGET-NORMALIZED COMPARISON

We now incorporate a compute budget $B$ and normalize both strategies by their per-trial costs $C_g$ and $C_\ell$.

**Theorem 1** (Compute-normalized advantage of LoTTS). *Given a total compute budget $B$, the expected quality gains of Global TTS and LoTTS satisfy*

$$\mathbb{E}[\Delta r]_{\text{global},B} \ \geq \ \frac{B}{C_g}\Big(s\,\theta_g\,\overline{\delta} - (M-s)\,h_g\,\overline{\gamma}\Big), \tag{14}$$

$$\mathbb{E}[\Delta r]_{\text{local},B} \ \geq \ \frac{B}{C_\ell}\Big(\rho\,s\,q\,\overline{\delta} - \rho\,s\big(\tfrac{1}{\pi}-1\big)h_\ell\,\overline{\gamma}\Big). \tag{15}$$

## G.6 DOMINANCE CONDITION

LoTTS outperforms Global TTS whenever

$$\frac{\rho}{C_\ell}\Big(q\overline{\delta} - (\tfrac{1}{\pi}-1)h_\ell\overline{\gamma}\Big) \ > \ \frac{1}{C_g}\Big(\theta_g\overline{\delta} - (\tfrac{M}{s}-1)h_g\overline{\gamma}\Big). \tag{16}$$

**What the inequality encodes.** Condition equation 16 compares the *expected per-compute improvement* of LoTTS (left) and Global TTS (right). On the left, $\rho$ (recall) scales how many defective patches LoTTS actually touches; $q\overline{\delta}$ is the average benefit of successfully repairing a defect; the subtraction $(\tfrac{1}{\pi}-1)h_\ell\overline{\gamma}$ accounts for the expected harm caused by *false positives* among the selected patches; and the entire effect is normalized by the localized trial cost $C_\ell$. On the right, Global TTS can in principle repair all $s$ defects with probability $\theta_g$, contributing $\theta_g\overline{\delta}$ on average, but it risks harming *every* clean patch; this produces the sparsity-amplified penalty $(\tfrac{M}{s}-1)h_g\overline{\gamma}$, because there are $(M-s)$ clean patches versus only $s$ defective ones. That global trade-off is normalized by the global trial cost $C_g$.

**Equivalent re-arrangement (solving for $\rho$).** For diagnostics, it is useful to isolate the recall $\rho$ required for LoTTS to dominate. Multiplying both sides of equation 16 by $C_\ell$ and dividing by the (assumed positive) bracket $q\overline{\delta} - (\tfrac{1}{\pi}-1)h_\ell\overline{\gamma}$ yields

$$\rho \ > \ \frac{C_\ell}{C_g}\,\frac{\theta_g\overline{\delta} - \big(\tfrac{M}{s}-1\big)h_g\overline{\gamma}}{q\overline{\delta} - (\tfrac{1}{\pi}-1)h_\ell\overline{\gamma}}, \qquad \text{provided} \quad q\overline{\delta} - (\tfrac{1}{\pi}-1)h_\ell\overline{\gamma} > 0. \tag{17}$$

The denominator in equation 17 is precisely the *net per-patch gain* for LoTTS when a patch is selected: it must be positive, otherwise false-positive harm outweighs the benefit of a true repair and localized editing should not be used.

**Minimal precision for the mask.** A convenient sufficient condition to ensure the denominator of equation 17 is positive is a lower bound on the precision $\pi$:

$$q\overline{\delta} - (\tfrac{1}{\pi}-1)h_\ell\overline{\gamma} \ > \ 0 \quad \Longleftrightarrow \quad \pi \ > \ \frac{1}{1 + \frac{q\overline{\delta}}{h_\ell\overline{\gamma}}}. \tag{18}$$

Intuitively, the more benign localized edits are (small $h_\ell\overline{\gamma}$) and/or the more effective they are (large $q\overline{\delta}$), the weaker the precision requirement on the mask.

**Special cases and insights.**

- **Benign-edit regime ($h_g = h_\ell = 0$).** Then equation 16 reduces to $\frac{\rho q}{C_\ell} > \frac{\theta_g}{C_g}$. LoTTS dominates if its recall-weighted success rate per unit compute exceeds that of global resampling.

- **Perfect mask ($\pi = \rho = 1$).** Condition equation 16 becomes $\frac{q}{C_\ell} > \frac{1}{C_g}\big(\theta_g - (\tfrac{M}{s}-1)\frac{h_g\overline{\gamma}}{\overline{\delta}}\big)$. Even if $C_\ell \approx C_g$, LoTTS still enjoys a *sparse-defect advantage*: the global harm term grows with $(M/s - 1)$ (many clean patches), whereas LoTTS avoids touching clean regions.

- **Equal costs** ($C_\ell = C_g$). The comparison is purely statistical: $\rho\big[q\overline{\delta} - (\frac{1}{\pi} - 1)h_\ell\overline{\gamma}\big] > \theta_g\overline{\delta} - \big(\frac{M}{s} - 1\big)h_g\overline{\gamma}$. LoTTS benefits from (i) nontrivial recall $\rho$, (ii) reasonable precision $\pi$ to keep FP harm small, and (iii) the fact that the global penalty scales with $(M/s - 1)$ when defects are sparse.

- **Connection to Best-of-$N$.** In many implementations, $\theta_g$ increases sublinearly with $N$ while $C_g$ grows roughly linearly, so $\theta_g/C_g$ quickly saturates. In contrast, LoTTS improves the *numerator* on the left of equation 16 by focusing on defect regions (larger effective $q\overline{\delta}$) and keeps the penalty term small by requiring only modest $\pi$ via equation 18.

**Practical reading.** Inequality equation 16 holds exactly when the *per-compute repair efficiency* of LoTTS—recall-weighted true repairs $q\overline{\delta}$ minus false-positive harm $(\frac{1}{\pi} - 1)h_\ell\overline{\gamma}$—exceeds that of Global TTS, whose harm term is amplified by the large number of clean patches $(M - s)$. Therefore, in the common *sparse-defect* regime ($s \ll M$) with a mask of moderate precision and recall, LoTTS is expected to dominate, even when $C_\ell$ is close to $C_g$.

### G.7 Corollaries

The dominance condition in Theorem 1 can be simplified under specific regimes of practical interest. These corollaries highlight intuitive scenarios where LoTTS has a clear advantage over global TTS.

**Corollary 1** (Benign edits). *If neither global nor localized resampling introduces harm ($h_g = h_\ell = 0$), then LoTTS dominates whenever*

$$\frac{\rho q}{C_\ell} > \frac{\theta_g}{C_g}.$$

**Interpretation.** In this special case, only successful repairs contribute to quality improvement, and all harm terms vanish. The comparison thus reduces to the relative efficiency of repairs per unit compute. LoTTS enjoys a recall-weighted success probability $\rho q$, while global TTS has success probability $\theta_g$ across all defects. After normalization by costs, LoTTS dominates whenever its recall-adjusted repair efficiency outweighs the global rate. This formalizes the intuition that even if local edits do not harm clean regions, they are more cost-effective so long as recall is not too low.

**Corollary 2** (Sparse-defect regime). *If the number of defects is much smaller than the total number of patches ($s \ll M$), with non-negligible harm probability $h_g > 0$ for global resampling but small $h_\ell$ for localized resampling, then LoTTS dominates under much weaker conditions.*

**Interpretation.** When defects are sparse, global resampling pays a high penalty because harm can occur in any of the $(M - s)$ clean patches. The term $(M - s)h_g\overline{\gamma}$ therefore dominates the global expectation, even if $\theta_g$ is large. By contrast, localized resampling only touches $\rho s/\pi$ patches, so the harm penalty scales with the number of false positives rather than the entire clean set. Thus in sparse-defect regimes, LoTTS achieves a strictly more favorable repair-to-harm tradeoff, and the condition for its dominance is considerably easier to satisfy.

### G.8 Relation to Best-of-N

A natural question is whether Global TTS can match or surpass LoTTS simply by increasing $N$, i.e., sampling more candidates and selecting the best. We now analyze the cost–benefit tradeoff of Best-of-$N$.

**Corollary 3** (Best-of-$N$ scaling). *For global Best-of-$N$, the compute cost and repair probability evolve as*

$$C_g(N) = N \cdot C_g(1), \qquad \theta_g(N) = 1 - \big(1 - \theta_g(1)\big)^N,$$

*where $C_g(1)$ and $\theta_g(1)$ denote the per-trial cost and repair probability for a* single *global sample. Thus, the budget-normalized efficiency is*

$$\frac{1}{NC_g(1)}\Big(\theta_g(N)\overline{\delta} - \big(\tfrac{M}{s} - 1\big)h_g\overline{\gamma}\Big).$$

This expression shows that $\theta_g(N)$ increases sublinearly in $N$ due to diminishing returns, while $C_g(N)$ grows linearly. As a result, beyond a moderate $N$, the efficiency of Best-of-$N$ saturates

or even decreases, since the harm penalty term remains proportional to the large number of clean patches $(M - s)$.

In contrast, LoTTS achieves improvements with trial cost $C_\ell$ and without paying a penalty that scales with all clean patches. Therefore, even when $N$ is large, LoTTS may still dominate in the sparse-defect regime, provided the mask has non-trivial recall and precision. This establishes that LoTTS is not merely equivalent to "Best-of-$N$ with smaller $N$," but can be fundamentally more efficient by focusing computation only on defective regions.

### G.9 TAKEAWAY

Theorem 1 provides a general, compute-normalized criterion under which LoTTS surpasses Global TTS. Two practically important regimes follow.

**(A) General regime ($C_\ell \ll C_g$).** When localized trials are cheaper than global trials ($C_\ell \ll C_g$), the dominance condition in equation 16 is typically easy to satisfy: LoTTS concentrates compute on defect-prone regions and avoids paying a harm penalty on all clean patches. Even with moderate mask quality (non-trivial precision $\pi$ and recall $\rho$), the left-hand side of equation 16 is boosted by the factor $1/C_\ell$, while the right-hand side suffers both from its larger denominator $C_g$ and from the sparsity-amplified harm term $\left(\frac{M}{s} - 1\right) h_g \overline{\gamma}$.

**(B) Our setting ($C_\ell \approx C_g$; same number of steps; full-image denoise but masked update).** In our implementation, local refinement uses the same number of denoising steps as the global sampler ($C_\ell \approx C_g$), and applies the reverse update only on masked pixels while unmasked pixels follow the scheduled Gaussian branch. In this equal-cost case, equation 16 simplifies to a purely statistical comparison:

$$\rho \left[ q\,\overline{\delta} - \left(\tfrac{1}{\pi} - 1\right) h_\ell\,\overline{\gamma} \right] \;>\; \theta_g\,\overline{\delta} - \left(\tfrac{M}{s} - 1\right) h_g\,\overline{\gamma}. \tag{19}$$

It says LoTTS wins exactly when its *recall-weighted net per-patch gain* (true repairs minus FP harm) exceeds the global method's net gain (true repairs minus widespread harm across all clean patches). Three concrete implications make equation 19 favorable to LoTTS in practice:

1. **Sparse-defect advantage.** If $s \ll M$, the global harm term on the right, $\left(\frac{M}{s} - 1\right) h_g \overline{\gamma}$, is magnified by the large number of clean patches. LoTTS, instead, pays harm only for false positives and thus scales with $\rho s(\frac{1}{\pi} - 1)$.

2. **Minimal precision requirement.** From equation 17, LoTTS needs the denominator $q\overline{\delta} - (\frac{1}{\pi} - 1)h_\ell\overline{\gamma}$ to be positive. This yields a precision threshold $\pi > \left(1 + q\overline{\delta}/(h_\ell\overline{\gamma})\right)^{-1}$ ( equation 18). If localized edits are relatively benign (small $h_\ell\overline{\gamma}$) and/or effective (large $q\overline{\delta}$), the required $\pi$ can be quite modest.

3. **Recall target.** Still in the equal-cost case, the minimal recall to dominate is

$$\rho \;>\; \frac{\theta_g\overline{\delta} - \left(\frac{M}{s} - 1\right) h_g\overline{\gamma}}{q\overline{\delta} - (\frac{1}{\pi} - 1)h_\ell\overline{\gamma}},$$

   cf. equation 17 with $C_\ell/C_g \approx 1$. Thus, as long as the mask has moderate recall and precision (exceeding the above thresholds), LoTTS will outperform Global TTS even when both consume the same number of steps.

**When LoTTS may not help.** If defects are *dense* ($s \approx M$), or the mask precision falls *below* the threshold in equation 18 so that FP harm overwhelms gains, or the per-defect local success $q$ is substantially *worse* than the global success $\theta_g$, then the right-hand side of equation 19 may dominate. These are precisely the regimes where full regeneration or Best-of-$N$ is a reasonable fallback.

**Bottom line.** In the common sparse-defect regime with a mask of modest quality (precision above equation 18 and recall above equation 17), LoTTS delivers higher expected quality gain per compute *even when $C_\ell \approx C_g$ (same steps)*, because it avoids the global harm that scales with the number of clean patches. The general Theorem 1 covers both $C_\ell \ll C_g$ and $C_\ell \approx C_g$; Our implementation corresponds to the latter as a special, yet still favorable, case.

## H   THE USE OF LARGE LANGUAGE MODELS

The use of Large Language Models (LLMs) in this work was only restricted to grammar check and minor editing. All conceptual development, experimental design, and analyses were conducted independently by the authors.

