# OpenReview forum: "Don’t Scale It All: Training-Free Localized Test-Time Scaling for Diffusion Models"
_ICLR.cc/2026/Conference — ICLR 2026 Conference Withdrawn Submission_

### Official Review · Reviewer_g4sA · 2025-10-24

**Soundness:** 3
**Presentation:** 3
**Contribution:** 2
**Rating:** 4
**Confidence:** 4

**Summary:**

This paper presents a TTS technique for fixing generation errors in diffusion models focusing on improving local regions. The main contribution is a two staged pipeline first by detecting regions of low quality and then by devising a local based refinement/resampling strategy

**Strengths:**

- The idea of identifying and fixing defective local regions is meaningful
- The methods looks pretty effective in terms of numeric results (Table 1) for a variety of diffusion models. It's clearly the best reported method in the table

**Weaknesses:**

- Although the idea is meaningful, the method basically constitutes applying a series of heuristics for both identifying the errors and then fixing them. I'm not sure this is of sufficient novelty for ICLR.
- It's also not very clear what kind of errors the proposed technique is trying to fix.

**Questions:**

- How quality is defined is not clear: what does it mean a “high-quality image”?  It sounds to me that this is vague. Also shouldn’t this definition be more fine-grained for example with respect to blur, artifacts, noise etc. etc.  Do you have some empirical evidence that this works?
- Shouldn’t Eq. 7 be multiplicative?
- How does the selection of r affects the results? What happens if r is too small or too big? Shouldn’t r be dynamically (per sample) selected? I think the analysis of L1055 seems thin
- How do you know that resampling will fix the issues? Aren’t there cases that consistency doesn’t work resulting in more artifacts?
- Not all mistakes are clear in Fig. 5. For example can you clarify the examples of the first row ?

---

### Official Review · Reviewer_6NTa · 2025-10-29

**Soundness:** 2
**Presentation:** 3
**Contribution:** 2
**Rating:** 4
**Confidence:** 4

**Summary:**

This paper introduces a localized test-time scaling for diffusion models. The proposed framework shifts the attention from global image refinement in previous works to artifact-prone local regions, improving the efficiency. It proposes prompt-driven noise search and defect-aware resampling to address the challenges introduced by localization. The framework excels in several benchmarks on 3 diffusion variants compared to previous baselines.

**Strengths:**

1. The localized test-time scaling is intuitive and promising for higher efficiency compared to the previous global image approaches.
2. The framework generally outperforms previous baseline approaches across three diffusion variants on human-aligned and automated metrics, demonstrating its potential for wide and broad usage.
3. The scaling efficiency compared to Best-of-N showcases the speedup.
4. Visualization clearly provides method framework details and the comparison of the proposed methods with baselines.

**Weaknesses:**

1. The qualitative improvement (especially Figure 5) is not very obvious. The "After" images are of similar quality to the "Before". More complex prompt images may help to demonstrate the effectiveness of the proposed method
2. The ablation study is a bit vague to compare the effects of different components. In Figure 6, and Section 5.2, it is not clearly stated whether the "w/o" component is deleted step by step or taken away separately, making it hard to validate the improvement introduced by each component.

**Questions:**

1. The statement that $CA_{diff} = CA_{neg}-CA_{pos}$ emphasizes artifact-prone spatial localization is a bit unclear. This formula can be viewed as a learning direction from bad masks ("low quality") to better areas ("high_quality"), where the model obtains the masks introduced from "low quality" prompt. For general inference time, when we provide the positive prompt always with "high quality", the question is whether the artifact-prone mask introduced by this formula still corresponds to the defect regions of generations from positive prompt.
2.  What is the search root in Figure 4? Although some generated examples of LoTTS in this Figure are better than previous baselines, the changes compared to the search root do not seem very obvious.
3. References about Resampling / Best-of-N to previous approaches would help for baseline tracking. Although there is more information in the supplementary, the details, such as the number of "N", are still not clear
4. Ablation Study (stated in weakness 2), It would be helpful to state clearly about the ablation configuration. For example, w/o $CA_{orig}'$ is taking away $CA_{orig}'$ but still with propogation or without.

---

### Official Review · Reviewer_CeNk · 2025-10-30

**Soundness:** 3
**Presentation:** 3
**Contribution:** 2
**Rating:** 4
**Confidence:** 2

**Summary:**

The work introduces an approach for zero-shot defect correction in generated images. The defects are first localized by comparing the cross-attention maps between a high and low quality prompt. Then, the masked regions are re-generated. The approach is tested with different backbones showing promising results.

**Strengths:**

- Very detailed explanation of the approach both in the method, but also well detailed in the appendix.
- Visualizations for both good and bad (failure cases) provided.
- Good results compared with the baselines
- Tested using different backbones
- The method is sound and interesting

**Weaknesses:**

- I am not very familiar with the sup-topic of defect correction, so it's a bit unclear to me what are the defect correction methods that this work compares with. I checked the ones cited in the results section, but they don't appear to be specialized tot this task. Instead, they are more general in nature, ensuring that the given model produces satisfactory results while closely following the prompt. Are there any methods explicitly designed for this (be it training-free or training aware) that this work compares with? If not, it will be good to have this added, as now it's a bit unclear how well this method fairs against specialized methods.

- Continuing on this, what rewards/selection criteria were used for the baseline to steer them on the defect correction direction?

- It's unclear to me if this work really fixes defects as opposed to better prompt alignment in general. The prompt itself mostly encourages the addition of more details/sharpens areas. In some cases I would argue, it even introduces some issues (see figure 10: the fridge reflection is unrealistic afterwards, the tree was sharpened and has inconsistent blur level, the pavement becomes less realistic etc).

- Similarly, the prompt chosen may interfere with a user intent of generating a more realistic/noisy image intentional. What other prompts were considered?

**Questions:**

- How does this approach scale to bigger and more diverse evaluation datasets, such as LAION?
- Is the current approach working equally well at both high and low resolution?
- What other alternative rewards (e.g: based on detection etc) where used as a comparison?
- Can the approach be combined with any other strategy?

---

### Official Review · Reviewer_fu1p · 2025-10-31

**Soundness:** 2
**Presentation:** 3
**Contribution:** 2
**Rating:** 4
**Confidence:** 4

**Summary:**

This paper presents LoTTS, a training-free localized test-time scaling (TTS) framework for diffusion and flow models. Rather than applying extra computation to the entire image, LoTTS (i) identifies local defects using attention-based, quality-aware masks that contrast “high-quality” and “low-quality” prompt responses, propagate them via self-attention, and reweight with the original prompt; and (ii) performs localized resampling at intermediate timesteps, followed by a short global refinement to ensure consistency. Experiments on SD2.1, SDXL, and FLUX demonstrate that LoTTS improves both human-aligned and automated quality metrics while reducing GPU cost by 2–4× compared to Best-of-N sampling.

**Strengths:**

(1)  LoTTS introduces a quality-aware local optimization strategy that allocates computational resources more efficiently than traditional global resampling approaches.

(2) Leveraging cross-attention maps from diffusion models to locate and refine low-quality regions offers a novel and insightful perspective on test-time optimization for generative models.

(3) The inclusion of pseudocode and theoretical analysis provides clarity and rigor, showing that LoTTS achieves higher expected quality gains under reasonable assumptions, thereby offering a solid theoretical foundation for the method.

**Weaknesses:**

(1) Lack of empirical evidence for spatial heterogeneity: The paper asserts that “test-time scaling is inefficient due to spatial quality heterogeneity,” but provides no quantitative or visual evidence to support this claim. It remains unclear how common or severe this heterogeneity is in typical diffusion outputs, making the argument more intuitive than empirically grounded.

(2) Unclear prompt selection mechanism: The paper mentions using high-quality and low-quality prompts to guide defect localization but lacks concrete details on how these prompts are designed or selected. Since image quality is not always directly inferable from textual descriptions, relying solely on prompts to detect poor-quality regions may be unreliable or subjective.

(3) Potential instability of cross-attention maps: The method identifies low-quality regions by contrasting cross-attention maps between high- and low-quality prompts. However, it does not address how to mitigate false positives caused by inherent generative inconsistencies, such as texture variation or background artifacts, which may not truly indicate quality degradation.

(4) Lack of clarity in consistency maintenance:  The section describing how LoTTS maintains global consistency during localized resampling would benefit from a clearer illustration. A framework showing the interplay between local and global refinement stages would help readers understand the mechanism more intuitively.

(5) Inaccurate characterization of prior work: The critique that “Best-of-N search wastes compute since it discards promising samples” oversimplifies the landscape of existing test-time scaling methods. Several prior works already incorporate compute reuse or guided re-ranking to improve efficiency. The paper should clarify what makes LoTTS uniquely efficient in this context, beyond its localized refinement strategy.

(6) The legend in Figure 1 is confusing. Specifically, the “selected” and “rejected” regions appear mislabeled. This inconsistency could mislead readers about the algorithm’s selection process and should be corrected for clarity.

**Questions:**

N/A

---

### Note · Authors · 2025-11-12

I have read and agree with the venue's withdrawal policy on behalf of myself and my co-authors.